# Rationally designed azobenzene photoswitches for efficient two-photon neuronal excitation

Gisela Cabré[1], Aida Garrido-Charles[2], Miquel Moreno[1], Miquel Bosch[2], Montserrat Porta-de-la-Riva[3], Michael Krieg[3], Marta Gascón-Moya[1], Núria Camarero[2], Ricard Gelabert[1], José M. Lluch [1], Félix Busqué[1], Jordi Hernando[1], Pau Gorostiza [2,4,5] & Ramon Alibés [1]

Manipulation of neuronal activity using two-photon excitation of azobenzene photoswitches with near-infrared light has been recently demonstrated, but their practical use in neuronal tissue to photostimulate individual neurons with three-dimensional precision has been hampered by firstly, the low efficacy and reliability of NIR-induced azobenzene photo-isomerization compared to one-photon excitation, and secondly, the short *cis* state lifetime of the two-photon responsive azo switches. Here we report the rational design based on theoretical calculations and the synthesis of azobenzene photoswitches endowed with both high two-photon absorption cross section and slow thermal back-isomerization. These compounds provide optimized and sustained two-photon neuronal stimulation both in light-scattering brain tissue and in *Caenorhabditis elegans* nematodes, displaying photoresponse intensities that are comparable to those achieved under one-photon excitation. This finding opens the way to use both genetically targeted and pharmacologically selective azobenzene photoswitches to dissect intact neuronal circuits in three dimensions.

[1] Departament de Química, Universitat Autònoma de Barcelona (UAB), Cerdanyola del Vallès 08193, Spain. [2] Institut de Bioenginyeria de Catalunya (IBEC), Barcelona Institute of Science and Technology (BIST), Barcelona 08028, Spain. [3] Institut de Ciències Fotòniques (ICFO), The Barcelona Institute of Science and Technology (BIST), Castelldefels, Barcelona 08860, Spain. [4] Institució Catalana de Recerca i Estudis Avançats (ICREA), Barcelona 08010, Spain. [5] Centro de Investigación Biomédica en Red en Bioingeniería, Biomateriales y Nanomedicina (CIBER-BBN), Zaragoza 50018, Spain. These authors contributed equally: G. Cabré, A. Garrido-Charles. Correspondence and requests for materials should be addressed to P.G. (email: pau@icrea.cat) or to R.A. (email: ramon.alibes@uab.cat)

A zobenzene photoswitches[1] are at the core of most recently developed strategies to manipulate biological functions with light[2–4], which among other systems, enable remote control of cell receptors and channels[5–7]. Toward these applications, much effort is being devoted to optimize the light-induced response of azobenzenes (e.g. long-wavelength operation[8–13] or modulation of *cis* state thermal stability[9,13]).

A crucial aspect that must ineluctably be addressed to unleash the full potential of azobenzene photoswitches is multiphoton excitation with near-infrared radiation (NIR, ~700–1400 nm)[14,15], which enables three-dimensional (3D) sub-micrometric resolution[16], deeper penetration into tissue with lower photodamage[17], and patterned illumination[18,19]. However, in contrast to the progress made with optogenetics[20,21], efficient multiphoton operation of azobenzenes still remains a challenge, mainly due to the low two-photon (2P) absorption cross sections ($\sigma_2$) of most of these compounds under NIR light excitation[22,23].

This is the case of **MAG**, the first-generation azobenzene-based photoswitchable tethered ligand (PTL) used for the preparation of light-gated glutamate receptors (LiGluR, Fig. 1a and Supplementary Fig. 1)[24,25]. Upon conjugation to a cysteine residue genetically engineered in kainate-type ionotropic glutamate receptor GluK2, **MAG** *trans-cis* photoisomerization allows optical control of ion channel opening and closing in LiGluR[24,25], a behavior profusely applied to the study of neurotransmission in vitro and in vivo[26–30]. While effective light-gating of LiGluR is achieved via regular one-photon (1P) absorption of ultraviolet-visible (UV-vis) radiation[24–30], multiphoton operation with NIR light is preferred to stimulate selected cells located deep into tissues at high spatiotemporal resolution[31,32]. Indeed, 2P switching at ~750–900 nm was recently demonstrated for LiGluR after functionalization with **MAG**[33] and its analogous ligand **MAG₀** (Supplementary Fig. 1)[34]; however, rather limited responses were obtained owing to the very low 2P absorption cross section of the symmetrically substituted azobenzene core of these compounds ($\sigma_2 = 10$ GM for *trans*-**MAG₀** at 820 nm[34], Fig. 1b).

Two different strategies have been explored to enhance the 2P activity of MAG-type PTLs for multiphoton LiGluR operation. On one hand, sensitized photoswitching with NIR light was attempted by tethering a 2P-absorbing antenna to the ligand[33,35], which however compromised its biological activity due to decreased solubility in water, low conjugation efficiency to cysteine-tagged GluK2, and/or reduced affinity toward the receptor-binding site[33,35]. On the other hand, electronically asymmetric azobenzenes were proposed to intrinsically increase their 2P absorption cross sections[22,23,36–38]. For MAG-type PTLs and other related compounds, this concept was examined by introducing a strong electron-donating amino group in the fourth position of their azobenzene core (e.g. in **MAG₂ₚ** and **MAG₄₆₀**, Fig. 1b and Supplementary Fig. 1)[33,34,39]. A notable increase in 2P absorption was observed in these cases ($\sigma_2 = 80$ GM for *trans*-**MAG₄₆₀** at 850 nm[34]), though at the expense of dramatically decreasing the thermal stability of the *cis* isomer of the photoswitch down to the sub-second timescale. As such, this prevented large photoresponses to be obtained for the 2P operation of LiGluR with **MAG₂ₚ** and **MAG₄₆₀**, since rapid thermal relaxation of the *cis* state of the switch impeded building up a large population of the open state of the ion channel[33,34].

Here we report the computationally based rational design and preparation of MAG-type photoswitches displaying both high 2P biological activity with NIR light and large *cis* isomer thermal lifetime ($\tau_{cis}$). Our approach toward this goal relies on the accurate selection of the substitution pattern of the azoaromatic core of the system, which should allow for electronic asymmetry (i.e. enhanced $\sigma_2$ in the NIR region) without compromising the

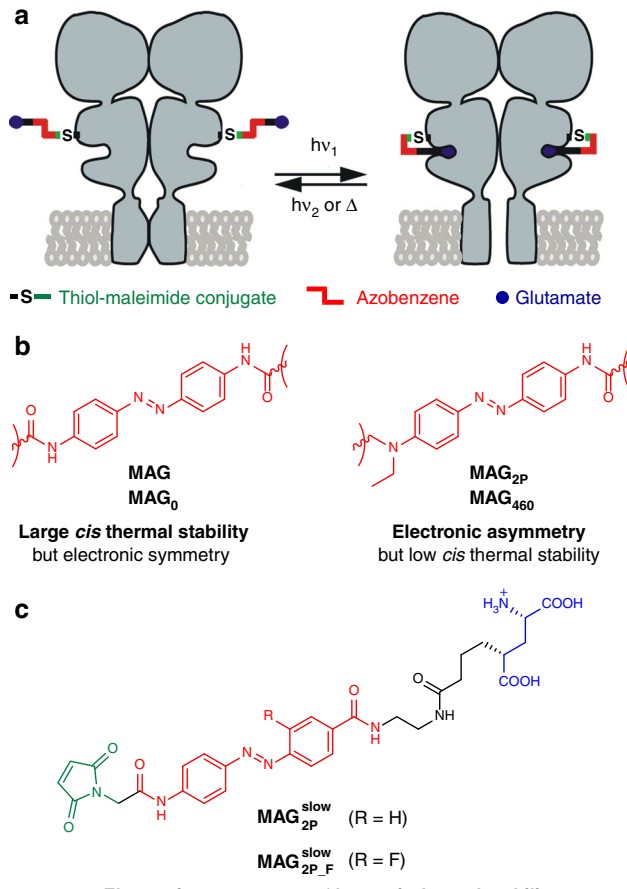

**Fig. 1** Strategy toward optimized azobenzene photoswitches for the two-photon (2P) excitation of light-gated glutamate receptors (LiGluR). **a** Operating mode of MAG-type photoswitchable tethered ligands (PTLs) on LiGluR, which are composed of three covalently tethered units: a glutamate ligand, an azobenzene core, and a maleimide group that binds to a cysteine residue genetically engineered in the receptor. Ultraviolet-visible (one-photon; 1P) or near-infrared (2P) light excitation induces glutamate recognition and channel opening via *trans → cis* isomerization, which results in ion flow across the membrane. This process is reverted by illumination with visible light excitation (1P) or thermal back-isomerization of the *cis* state of the switch. **b** Structures of the azobenzene cores of **MAG**, **MAG₀**, **MAG₂ₚ**, and **MAG₄₆₀** PTLs proposed for the photoswitching of LiGluR under 1P and 2P excitation conditions. **c** Structures of PTLs **MAG₂ₚˢˡᵒʷ** and **MAG₂ₚ_Fˢˡᵒʷ**

thermal stability of its *cis* state. Importantly, this strategy could be expanded to other azobenzene-based photoswitches for tailoring their response under multiphoton excitation.

## Results

**Design and synthesis of high 2P-responsive MAG-type PTLs.** Although the most efficient manner to increase the 2P absorption cross section of azobenzenes lies in the push-pull substitution of their aromatic core[22,23], the introduction of strong mesomeric electron-donating (EDG), and/or electron-withdrawing (EWG) groups concomitantly accelerates their *cis → trans* thermal back-isomerization in the dark (e.g. with EDG = 4-NR₂ and EWG = 4-NO₂)[1]. A compromise must therefore be met to obtain azo derivatives with both large $\sigma_2$ and $\tau_{cis}$ values. In order to rationally devise these compounds, we computed the 2P absorption cross section and the *cis* state thermal stability for a series of model azoaromatic photochromes using the time-dependent density

**Table 1 2P absorption cross sections and *cis* thermal stabilities of model azo compounds in water**

| Azo compound[a] | $\sigma_{2,trans}$ (GM)[b] | $\Delta E^{\ddagger}_{cis-trans}$ (kJ mol$^{-1}$)[c] |
|---|---|---|
| **Azo$^{MAG}$** | | |
| R = 4-NHCOMe | 0[d] | 91.2 (25.5 min)[f] |
| R′ = 4′-NHCOMe | | |
| **Azo$^{MAG2p}$** | | |
| R = 4-NHCOMe | 56[e] | 67.7 (118 ms)[f] |
| R′ = 4′-NMe$_2$ | | |
| **Azo1** | | |
| R = 4-NHCOMe | 58 | 97.2 (4.9 h) |
| R′ = 4′-CONH$_2$ | | |
| **Azo2** | | |
| R = 4-NHCOMe | 69 | 98.1 (7.0 h) |
| R$_1$′ = 4′-CONH$_2$ | | |
| R$_2$′ = 2′-F | | |
| **Azo3** | | |
| R = 4-NHCOMe | 82 | 101.0 (22.6 h) |
| R$_1$′ = 4′-CONH$_2$ | | |
| R$_2$′ = 2′,4′-F | | |
| **Azo4** | | |
| R = 4-NHCOMe | 112 | 46.0 (18.6 μs) |
| R$_1$′ = 4′-CONH$_2$ | | |
| R$_2$′ = 2′-NO$_2$ | | |
| **Azo5** | | |
| R = 4-NHCOMe | 51 | 112.0 (80 days) |
| R$_1$′ = 4′-CONH$_2$ | | |
| R$_2$ = R$_2$′ = 2′,4′-F | | |

Calculations performed at the CAM-B3LYP/6–31G(d) level and accounting for solvent (water) effects with a self-consistent PCM continuum method

*2P* two-photon, *PCM* polarizable continuum model

[a]Structures shown in Supplementary Fig. 2

[b]2P absorption cross section for the $S_0 \rightarrow S_2$ transition of the *trans* isomer

[c]In all the cases, the lowest-energy barrier height for the thermal *cis* → *trans* isomerization was found to correspond to an inversion mechanism. In parentheses $\tau_{cis}$ values estimated at 298 K from Eyring equation are shown

[d]$\sigma_2 = 10$ GM[34] for *trans*-**MAG$_0$** containing an **Azo$^{MAG}$** core

[e]$\sigma_2 = 80$ GM[34] for *trans*-**MAG$_{460}$** containing an **Azo$^{MAG2p}$** core

[f]Estimated from the experimental $\tau_{cis}$ values reported for **MAG**[25] and **MAG$_{2P}$**[33] at room temperature using Eyring equation

functional theory (TDDFT, Table 1, Supplementary Fig. 2, and Supplementary Tables 1 and 2). The azobenzene cores of **MAG/MAG$_0$** (**Azo$^{MAG}$**, R = R′ = 4-NHCOMe) and **MAG$_{2P}$/MAG$_{460}$** (**Azo$^{MAG2p}$**, R = 4-NHCOMe, R′ = 4′-NMe$_2$) were taken as reference systems in these calculations, while several alternative azo compounds (**Azo1–Azo3**) were explored on the basis of two main design principles: (a) a push–pull substitution pattern to enhance $\sigma_2$ with respect to **MAG** and **MAG$_0$**; and (b) the use of weak mesomeric EDG (R = 4-NHCOMe) and EWG (R′ = 4′-CONH$_2$) as well as a strong inductive EWG (R′ = 2′-F and 2′,4′-F) to increase $\tau_{cis}$ relative to **MAG$_{2P}$** and **MAG$_{460}$**. For comparison purposes, two other model cases were considered: (a) an azo group alternatively bearing a strong mesomeric EWG substituent (**Azo4**, R′ = 2′-NO$_2$); and (b) an azo core with strong inductive EWGs on both aryl rings (**Azo5**, R = R′ = 2,4-F), a substitution pattern that enables long-wavelength 1P isomerization of azobenzenes[9] and was recently reported to allow for 2P operation with NIR light in biological samples[40].

As shown in Table 1 and Supplementary Tables 1 and 2, null 2P absorption was predicted for both the $S_0 \rightarrow S_1$ and $S_0 \rightarrow S_2$ transitions of *trans*-**Azo$^{MAG}$**, which is in good agreement with the behavior expected for purely centrosymmetric azobenzenes[22,23] and the minimal $\sigma_{2,trans}$ value experimentally determined for **MAG$_0$**[34]. By contrast, a major $\sigma_2$ value was computed for the $S_0 \rightarrow S_2$ band of electronically asymmetric *trans*-**Azo$^{MAG2p}$**, which is consistent with that measured for *trans*-**MAG$_{460}$**[34]. Interestingly, larger 2P absorption cross sections were calculated for model compounds *trans*-**Azo1–Azo4**, which showed a clear

dependence on the electronic asymmetry of their azo core: selective introduction of *o*-fluoro and *o*-nitro EWGs in one of the azobenzene aryl rings of *trans*-**Azo2–Azo4** led to a noteworthy increase in $\sigma_2$ with respect to *o*-unsubstituted *trans*-**Azo1**. On the contrary, electronic symmetrization of the azobenzene chromophore in *trans*-**Azo5** bearing four *o*-fluoro substituents inhibited this effect and even resulted in a slight decrease of the 2P absorption cross section with respect to *trans*-**Azo$^{MAG2p}$**. More importantly, the enhancement in $\sigma_{2,trans}$ observed for **Azo1–Azo3** was not found to detrimentally affect the stability of their *cis* isomer and rather large $\tau_{cis}$ values were predicted for these compounds in water, in contrast to the behavior observed for **Azo$^{MAG2p}$** in *cis*-**MAG$_{2P}$**/*cis*-**MAG$_{460}$** and *cis*-**Azo4** bearing a nitro group. This clearly demonstrates the advantages of designing electronically asymmetric azoaromatic photochromes with weak mesomeric EDG and EWG as well as strong inductive EWG, which emerge as ideal candidates for the preparation of azobenzene-based switches with high 2P activity for biological applications.

Because of the optimal $\sigma_{2,trans}$ and $\tau_{cis}$ values computed for **Azo1–Azo3**, the 2P absorption properties of the *cis* isomers of these compounds were theoretically predicted (Supplementary Tables 1 and 2), since they also affect the efficiency of the photoisomerization process of azobenzenes. On one hand, larger $\sigma_{2,trans}/\sigma_{2,cis}$ ratios were calculated for **Azo1–Azo3** with respect to **Azo$^{MAG}$** and **Azo$^{MAG2p}$** for their 2P-allowed $S_0 \rightarrow S_2$ transition. On the other hand, different excitation energies were found for both isomers of **Azo1–Azo3** as to favor selective 2P excitation of their *trans* states, although this behavior was observed to decrease with the number of fluorine substituents introduced. This, in combination with their high $\sigma_{2,trans}$ and $\tau_{cis}$ values and assuming similar *trans-cis* and *cis-trans* photoisomerization quantum yields[34], should result in a large 2P activity for *trans*-**Azo1–Azo3** in comparison to the *trans*-azoaromatic cores of **MAG/MAG$_0$** and **MAG$_{2P}$/MAG$_{460}$**.

Encouraged by our theoretical results on *trans*-**Azo1–Azo3**, two MAG-type PTLs were designed (**MAG$_{2P}^{slow}$** and **MAG$_{2P\_F}^{slow}$**, Fig. 1c). To favor synthetic accessibility, selective excitation of the *trans* isomer, as well as structural resemblance with **MAG** and, as such, replication of its light-gated control of LiGluR, we: (a) took the azo chromophores of **Azo1** and **Azo2** as models for the preparation of **MAG$_{2P}^{slow}$** and **MAG$_{2P\_F}^{slow}$**, and (b) employed similar linkers to those used in **MAG** to tether the maleimide and glutamate units to the azobenzene core.

The preparation of **MAG$_{2P}^{slow}$** (R = H) and **MAG$_{2P\_F}^{slow}$** (R = F) was achieved via a linear sequence where glutamate and maleimide functions were sequentially introduced into the azoaromatic photochrome of choice (Fig. 2)[24,25,33,35,41]. In this case, however, the azobenzene cores of **MAG$_{2P}^{slow}$** and **MAG$_{2P\_F}^{slow}$** were not commercially available and they had to be previously synthesized by diazotization of aminobenzoic acids **4a** (R = H) and **4b** (R = F) followed by coupling with sodium phenylamino-methanesulfonate **1**, which was readily available from aniline and the formaldehyde sodium bisulfite adduct[42]. Posterior removal of the amino protecting group delivered amino acids **5a** and **5b** in moderate yields. Then, the sequence continued by joining the monoprotected ethyldiamino tether to **5a** and **5b** using the carbodiimide coupling reagent *N*-ethyl-*N*′-(3-dimethyldiamino-propyl)-carbodiimide HCl along with 1-hydroxybenzotriazole hydrate and diisopropylethylamine as a base. Subsequent acid removal of the *tert*-butyl carbamate protection and introduction of previously described glutamate derivative **2**[35] using the same coupling conditions afforded intermediates **6a** and **6b** in 47 and 49%, respectively, for the three steps. The next reaction between these intermediates and the freshly prepared acid chloride of

**Fig. 2** Synthesis of $MAG_{2P}^{slow}$ and $MAG_{2P\_F}^{slow}$. Reagents and conditions: (a) 5.5 M HCl, $NaNO_2$; (b) (i) **1**, 0.86 M NaOAc; (ii) 1 M NaOH; (iii) 5.5 M HCl (38%, over the two steps, for **5a**, 47% for **5b**); (c) *tert*-butyl (2-aminoethyl)carbamate, *N*-ethyl-*N'*-(3-dimethyldiaminopropyl)-carbodiimide HCl (EDCI), 1-hydroxybenzotriazole hydrate (HOBt), diisopropylethylamine (DIPEA), THF; (d) 37% HCl, MeOH; (e) **2**, EDCI, HOBt, DIPEA, THF (47%, over the three steps, for **6a**, 49% for **6b**); (f) (i) **3**, ClCOCOCl, $CH_2Cl_2$, DMF; (ii) DIPEA, THF (86% for **7a**, 84% for **7b**); (g) trifluoroacetic acid (TFA), $CH_2Cl_2$ (80% for $MAG_{2P\_F}^{slow}$, quantitative yield for $MAG_{2P\_F}^{slow}$)

maleimide derivative **3**[43] furnished compounds **7a** and **7b**, which feature all the envisioned functional fragments of the final photoswitches. Finally, acid removal of all the protecting groups of the glutamate moiety delivered the target compounds $MAG_{2P}^{slow}$ and $MAG_{2P\_F}^{slow}$ as their monotrifluoroacetate salts in good yields.

Figure 3a shows the 1P absorption spectra of *trans*-$MAG_{2P}^{slow}$ and *trans*-$MAG_{2P\_F}^{slow}$ in aqueous buffer, which are compared to those of *trans*-**MAG** and *trans*-$MAG_{2P}$. As predicted by TDDFT calculations (Supplementary Table 2), very similar absorption signals were registered for the azoaromatic cores of *trans*-**MAG**, *trans*-$MAG_{2P}^{slow}$, and *trans*-$MAG_{2P\_F}^{slow}$, which are typical for azobenzene-type switches in the absence of strong mesomeric EDG and EWG[1]: they show an intense absorption band at $\lambda_{max} \sim$ 360 nm corresponding to the allowed 1P $S_0 \rightarrow S_2$ transition ($\pi\pi^*$ band), and a broad shoulder at $\lambda \sim$ 400–500 nm arising from the forbidden 1P $S_0 \rightarrow S_1$ transition ($n\pi^*$ band). This is in contrast with the absorption spectrum of *trans*-$MAG_{2P}$ bearing an amino-substituted azo core, the $\pi\pi^*$ band of which notably red-shifts and overlaps with the $n\pi^*$ band[33].

Upon excitation of *trans*-$MAG_{2P}^{slow}$ and *trans*-$MAG_{2P\_F}^{slow}$ $\pi\pi^*$ band at 365 nm in either organic (dimethylsulfoxide (DMSO)) or aqueous (99% phosphate-buffered solution (PBS):1% DMSO) media, spectral changes were observed in absorption that were consistent with 1P

$trans \rightarrow cis$ photoisomerization, as confirmed by [1]H nuclear magnetic resonance (NMR) (Supplementary Figs 3–4). High $trans \rightarrow cis$ photoisomerization quantum yields and *cis*-enriched photostationary states were determined for both $MAG_{2P}^{slow}$ and $MAG_{2P\_F}^{slow}$ at these irradiation conditions ($\Phi_{trans \rightarrow cis} \sim 0.15$ and % $cis^{PSS} \sim 70\%$ in aqueous buffer), which were similar to those measured for their azoaromatic cores and **MAG** (Supplementary Table 3). UV-vis absorption spectroscopy and [1]H NMR also revealed efficient 1P $cis \rightarrow trans$ photoisomerization of $MAG_{2P}^{slow}$ and $MAG_{2P\_F}^{slow}$ upon excitation of their $n\pi^*$ band at 473 nm ($\Phi_{cis \rightarrow trans} \sim 0.26$ and %$trans^{PSS} \sim 85\%$ in aqueous buffer), in analogy to the behavior registered for their azobenzene units and **MAG** (Supplementary Fig. 5 and Supplementary Table 4). In addition, thermal back-isomerization of *cis*-$MAG_{2P}^{slow}$ and *cis*-$MAG_{2P\_F}^{slow}$ was also observed at room temperature, which occurred in the time span of tens of minutes in aqueous media ($\tau_{cis} \sim 10$ min in 99% PBS:1% DMSO, Supplementary Fig. 6 and Supplementary Table 5). Importantly, this demonstrates that the azoaromatic cores selected for both $MAG_{2P}^{slow}$ and $MAG_{2P\_F}^{slow}$ possess long-lived *cis* isomers, as anticipated by our theoretical calculations. Therefore, these PTLs fairly reproduce the large $\tau_{cis}$ value of **MAG** and clearly surpass the thermal stability of *cis*-$MAG_{2P}$ and *cis*-$MAG_{460}$ despite

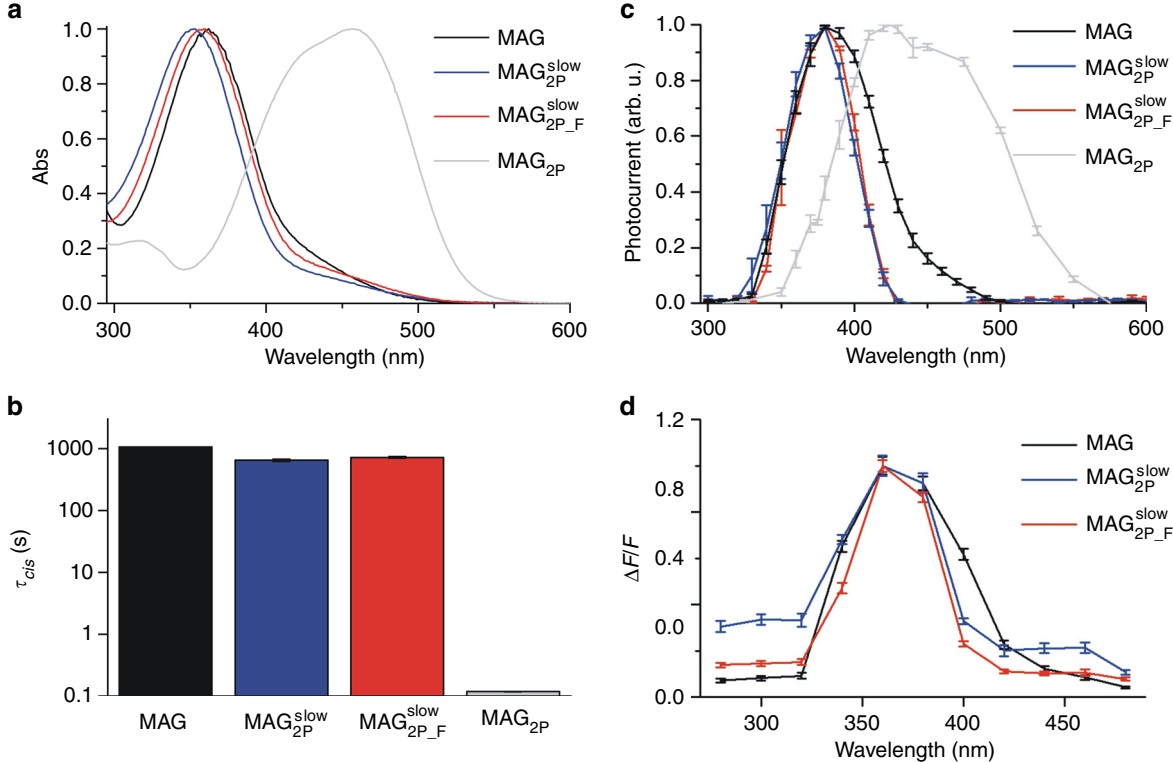

**Fig. 3** Photochemical and physiological characterization of $MAG_{2P}^{slow}$ and $MAG_{2P\_F}^{slow}$ under one-photon (1P) stimulation. **a** Normalized absorption spectra of *trans*-**MAG**, *trans*-$MAG_{2P}^{slow}$, *trans*-$MAG_{2P\_F}^{slow}$, and *trans*-$MAG_{2P}$ in 99% phosphate-buffered solution (PBS):1% dimethylsulfoxide (DMSO). **b** Thermal lifetimes of *cis*-**MAG**, *cis*-$MAG_{2P}^{slow}$, *cis*-$MAG_{2P\_F}^{slow}$, and *cis*-$MAG_{2P}$ at room temperature in 99% PBS:1% DMSO. Errors from the monoexponentials fits to obtain $\tau_{cis}$ are shown. **c** Normalized 1P action spectra recorded using whole-cell patch-clamp in human embryonic kidney 293 (HEK293) cells expressing GluK2-L439C after conjugation to **MAG**, $MAG_{2P}^{slow}$, $MAG_{2P\_F}^{slow}$, and $MAG_{2P}$ ($n = 5$, 7, 3, and 8 biologically independent cells, respectively). Errors are standard error of the mean (SEM). **d** Normalized 1P action spectra recorded using calcium imaging in HEK293 cells co-expressing GluK2-L439C and GCaMP6s after conjugation to **MAG**, $MAG_{2P}^{slow}$, and $MAG_{2P\_F}^{slow}$ ($n = 33$, 20, and 25 biologically independent cells, respectively). Errors are SEM. In **c** and **d** wavelength-dependent photoresponses were normalized to the maximum signal along the spectral range measured for each cell before averaging over different cells. Source data for **c** and **d** are provided as a source Data file

the push-pull substitution pattern of their azoaromatic photochromes (Fig. 3b).

In view of the optimal photochemical behavior established for $MAG_{2P}^{slow}$ and $MAG_{2P\_F}^{slow}$, we next explored their capacity to photocontrol LiGluR channels in living cells under 1P stimulation. Thus, GluK2 receptors bearing a cysteine residue at position L439C (GluK2-L439C) were expressed in human embryonic kidney 293 (HEK293) cells and incubated with the azobenzene-based compound of choice for selective conjugation ($MAG_{2P}^{slow}$, $MAG_{2P\_F}^{slow}$, **MAG**, and $MAG_{2P}$).

In a first set of experiments, the photoinduced operation of the resulting LiGluRs was evaluated using whole-cell patch-clamp, a technique that allows measuring the currents elicited across the cell membrane when modulating ion fluxes via channel opening and closing[24,25]. Large and repetitive electrophysiological signals were recorded in all the cases upon *trans* → *cis* photoisomerization of the PTLs (i.e. LiGluR channel opening) with UV and violet light ($MAG_{2P}^{slow}$, $MAG_{2P\_F}^{slow}$, and **MAG**) or broadband visible light ($MAG_{2P}$). Cell basal current levels could be next recovered by reverting back this process (i.e. LiGluR channel closing) upon illumination with blue and green radiation ($MAG_{2P}^{slow}$, $MAG_{2P\_F}^{slow}$, and **MAG**) or thermally in the dark ($MAG_{2P}$; Supplementary Fig. 7). By scanning the excitation wavelength used to induce LiGluR channel opening, the action spectra could be measured for each of the photoswitchable compounds under analysis (Fig. 3c). As expected from their photochemical properties, a very

similar spectral response was registered for $MAG_{2P}^{slow}$, $MAG_{2P\_F}^{slow}$, and **MAG**, which generated maximal electrophysiological signals when irradiating LiGluR-expressing cells at ~375 nm. By contrast, a broader red-shifted action spectrum was measured for $MAG_{2P}$ peaking at ~425 nm. In addition, further electrophysiological measurements were conducted to demonstrate that the photoswitches $MAG_{2P}^{slow}$ and $MAG_{2P\_F}^{slow}$: (a) preserve their long thermal *cis* lifetimes after tethering to GluK2 (~10 min, Supplementary Fig. 8) as well as the fast channel opening and closing times (<0.5 s, Supplementary Fig. 9 and Supplementary Table 6) previously reported for **MAG**[24,25]; and (b) do not inhibit the native physiological activity of the receptor, which retains their intrinsic response to free glutamate (Supplementary Fig. 10) and rapid desensitization kinetics (Supplementary Fig. 11) after photoswitch conjugation.

Taking advantage of the Ca$^{2+}$ permeability of LiGluR channels, their light-gated operation under 1P stimulation was further quantified by means of calcium imaging, using GCaMP6s co-expressed with GluK2-L439C in HEK293 cells. GCaMP6s is a genetically encoded intracellular fluorescent calcium indicator that undergoes a large increase in emission upon calcium ion complexation[44]. In this case, we focused on $MAG_{2P}^{slow}$-, $MAG_{2P\_F}^{slow}$-, and **MAG**-tethered LiGluRs, since the low thermal stability of *cis*-$MAG_{2P}$ is reported to lead to poor photoinduced calcium imaging signals[33]. In contrast, the large $\tau_{cis}$ values of $MAG_{2P}^{slow}$, $MAG_{2P\_F}^{slow}$, and **MAG** yielded intense, reversible, and

reproducible light-triggered fluorescent responses in GCaMP6s-expressing cells upon repetitive illumination with sequential pulses of UV-violet and green radiation (Supplementary Fig. 12). Similar action spectra were again measured for these PTLs using calcium imaging (Fig. 3d), and the maximal responses obtained for the three compounds at 360 nm were nearly equivalent (Supplementary Fig. 13). This, together with the whole-cell patch-clamp measurements conducted, proves that the 1P biological activity of **MAG** in LiGluRs is preserved for **MAG$_{2P}^{slow}$** and **MAG$_{2P\_F}^{slow}$** photoswitchable ligands bearing electronically asymmetric azoaromatic cores and slightly different structures.

**2P stimulation in cultured cells**. NIR-induced 2P operation of LiGluR with PTLs **MAG$_{2P}^{slow}$** and **MAG$_{2P\_F}^{slow}$** was studied by calcium imaging, since this enabled all-optical control and monitoring of the light-gated ion channels. Experiments were conducted in a confocal fluorescence microscope equipped with both continuous-wave visible lasers and a femtosecond pulsed Ti: Sapphire laser. This allowed for sequential and independent 1P and 2P stimulation of cells by raster-scanning the focused laser beam of choice over the sample while detecting the fluorescence signal of the calcium ion indicator for the whole field of view.

To evaluate the 2P physiological activity of **MAG$_{2P}^{slow}$** and **MAG$_{2P\_F}^{slow}$**, we first undertook experiments on HEK293 cells expressing both GluK2-L439C and R-GECO1[45], a red genetically encoded fluorescent Ca$^{2+}$ probe. Figure 4a depicts calcium imaging fluorescence responses measured for these cells after conjugation with **MAG**, **MAG$_{2P}^{slow}$**, and **MAG$_{2P\_F}^{slow}$** (see also Supplementary Movies 1-2). For comparison, all the cells were subjected to two consecutive cycles of 1P LiGluR photostimulation with violet light (405 nm) followed by two cycles of 2P excitation with NIR radiation (780 nm). After every stimulation, LiGluR was deactivated by 1P absorption of green light (514 nm). In averaged recordings over $n > 10$ cells in the field of view, no or very low signals were observed for **MAG**-tethered LiGluR upon irradiation with NIR light, which highlights the poor efficiency of 2P *trans → cis* photoisomerization for this ligand, as already reported[33,34]. In agreement with our theoretical calculations, a remarkable increase in 2P responses was observed when replacing **MAG** with **MAG$_{2P}^{slow}$** and **MAG$_{2P\_F}^{slow}$** bearing push-pull azoaromatic cores. In fact, 1P- and 2P-induced calcium imaging responses of similar intensities were measured with **MAG$_{2P\_F}^{slow}$**, for which the highest $\sigma_2$ value was anticipated owing to the high electronic asymmetry of its *trans*-azobenzene photochrome. Notably, such efficient 2P stimulation of LiGluR upon conjugation to **MAG$_{2P\_F}^{slow}$** was found to be reproducible, repetitive, and to occur with minimal photodegradation, since negligible variation in calcium imaging response was observed after four consecutive light-gating cycles for different individual cells (Fig. 4b).

By comparing the 1P- and 2P-induced calcium imaging responses recorded on the same cells (2P/1P ratio), a detailed assessment of the high multiphoton physiological activity of **MAG$_{2P}^{slow}$** and **MAG$_{2P\_F}^{slow}$** was performed. First, we determined the 2P action spectra of these compounds and **MAG** when conjugated to HEK293 cells co-expressing GluK2-L439C and R-GECO1, which showed a similar spectral distribution (Fig. 5a). In all the cases, rather broad 2P action spectra were found with maxima at ~780 nm, which reasonably agrees with twice the wavelength of the 1P absorption band of the 2P allowed $S_0 \rightarrow S_2$ transition of the *trans*-azobenzene core of these compounds (~360 nm). However, much higher 2P/1P ratios were obtained for **MAG$_{2P}^{slow}$** and **MAG$_{2P\_F}^{slow}$** compared to **MAG** at equivalent excitation conditions. In particular, when considering 2P

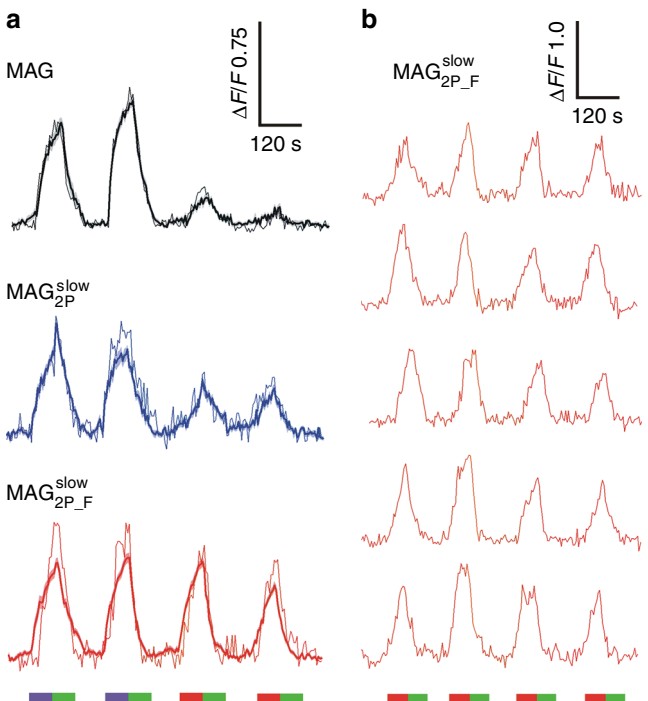

**Fig. 4** One-photon (1P) and two-photon (2P) stimulation of **MAG**, **MAG$_{2P}^{slow}$**, and **MAG$_{2P\_F}^{slow}$** in cultured cells. **a** Individual (thin lines) and average (thick lines) calcium imaging fluorescence traces recorded for human embryonic kidney 293 (HEK293) cells co-expressing GluK2-L439C and R-GECO1 after conjugation to **MAG** ($n = 16$ biologically independent cells), **MAG$_{2P}^{slow}$** ($n = 14$ biologically independent cells), and **MAG$_{2P\_F}^{slow}$** ($n = 34$ biologically independent cells). The bands around average traces plot the corresponding SEM. Both 1P (violet, 405 nm, power density = 0.37 mW μm$^{-2}$) and 2P excitation scans (red, 780 nm, power density = 2.8 mW μm$^{-2}$) were applied to open LiGluR channels and trigger calcium-induced R-GECO1 fluorescence enhancement, while 1P excitation scans (green, 514 nm, power density = 0.35 mW μm$^{-2}$) were applied to revert back the process. **b** Repetitive 2P-induced calcium imaging fluorescence responses recorded in five different HEK293 cells co-expressing GluK2-L439C and R-GECO1 after conjugation to **MAG$_{2P\_F}^{slow}$**. Source data for **a** are provided as a source Data file

stimulation of LiGluR at the spectral maximum (780 nm) and averaging over a large number of cells ($n > 25$), 3.5- and 6-fold increases in 2P/1P intensity ratio were measured for **MAG$_{2P}^{slow}$** and **MAG$_{2P\_F}^{slow}$** relative to **MAG**, respectively (Fig. 5b). Even more importantly, 2P stimulation of **MAG$_{2P}^{slow}$**- and **MAG$_{2P\_F}^{slow}$**-tethered LiGluR was observed for all the cells analyzed, while no multiphoton response could be measured for ~30% of the GluK2-L439C-expressing cells conjugated with **MAG** under the same illumination conditions (Fig. 5c). Overall, these results indicate that **MAG$_{2P}^{slow}$** and, especially, **MAG$_{2P\_F}^{slow}$** are very advantageous photoswitches to efficiently and reliably control LiGluR in neurotransmission studies under 2P excitation with NIR radiation.

**2P stimulation in hippocampal organotypic slices**. In view of the superior 2P stimulation performance observed for **MAG$_{2P\_F}^{slow}$** in GluK2-L439C-expressing HEK cells, we explored the use of this PTL to the multiphoton control with NIR light of neuronal cells embedded in their physiological environment. In this way we could not only assess the efficiency of **MAG$_{2P\_F}^{slow}$** in neurons with mature synapses naturally containing all endogenous glutamate

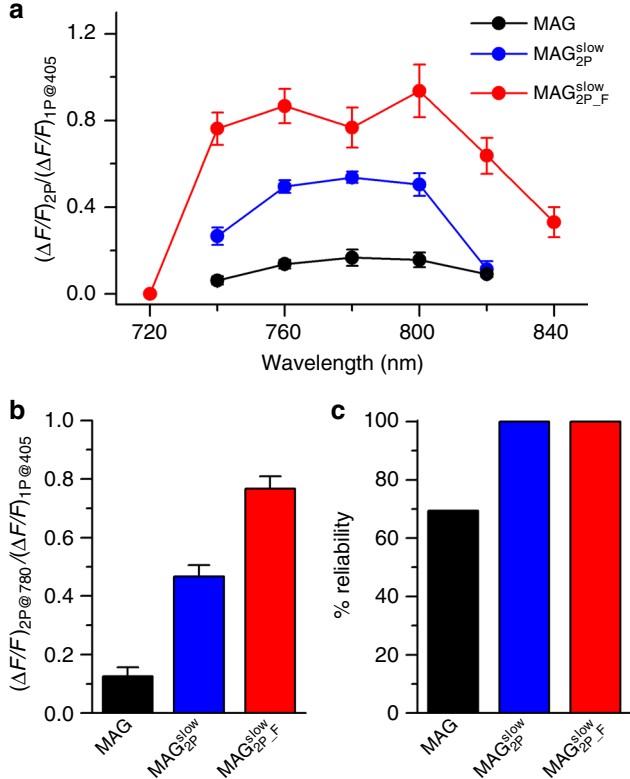

**Fig. 5** Average two-photon (2P) activity of **MAG**, **MAG**$_{2P}^{slow}$, and **MAG**$_{2P\_F}^{slow}$ in cultured cells. **a** 2P action spectra of **MAG**, **MAG**$_{2P}^{slow}$, and **MAG**$_{2P\_F}^{slow}$ after conjugation to GluK2-L439C-expressing human embryonic kidney 293 (HEK293) cells. Fluorescence calcium responses were measured using R-GECO1. Before averaging over different cells, the 2P responses of each cell were normalized with respect to the 1P response at 405 nm (**MAG**: 740, 760, 780, 800, and 820 nm; $n = 28, 33, 40, 7$, and 12 biologically independent cells, respectively; **MAG**$_{2P}^{slow}$: 740, 760, 780, 800, and 820 nm; $n = 20, 9, 12, 16$, and 17 biologically independent cells, respectively; and **MAG**$_{2P\_F}^{slow}$: 720, 740, 760, 780, 800, 820, and 840 nm; $n = 14, 17, 18, 86, 15, 23$, and 17 biologically independent cells, respectively). Errors are SEM. **b** Ratio between the 2P and 1P responses of **MAG**, **MAG**$_{2P}^{slow}$, and **MAG**$_{2P\_F}^{slow}$ for the same cells excited at 780 and 405 nm, respectively. Errors are SEM. **c** Reliability of the 2P calcium imaging response elicited in GluK2-L439C-expressing HEK293 cells after conjugation to **MAG**, **MAG**$_{2P}^{slow}$, and **MAG**$_{2P\_F}^{slow}$ ($n = 72, 25$, and 86 biologically independent cells, respectively). Reliability is expressed as the percentage of transfected cells showing measurable 2P stimulation signals. Source data for **a** and **b** are provided as a source Data file

receptors but also test the ultimate advantages of 2P stimulation. With this aim, we prepared organotypic slice cultures from neonatal rat hippocampi and biolistically transfected them with GluK2-L439C-eGFP and RCaMP2[46]. This allowed the cells expressing LiGluR within the slices to be localized by the green fluorescence of enhanced green fluorescent protein (eGFP), while simultaneously recording their activity after 1P or 2P stimulation by monitoring the red fluorescence of the calcium ion probe RCaMP2 (Fig. 6a). Transfected hippocampal slices were finally incubated with **MAG** or **MAG**$_{2P\_F}^{slow}$ and their photoresponses after consecutive 1P and 2P stimulation were measured and compared. Neither Concanavalin A nor toxins were used in these experiments to inhibit GluK2 desensitization upon prolonged binding to the glutamate unit of *trans*-**MAG** and *trans*-**MAG**$_{2P\_F}^{slow}$. Thus, our measurements on brain slices truly reported on intact neuronal gating and connectivity.

As depicted in Fig. 6b, d, slices incubated with **MAG** showed robust photoresponses during 1P stimulation (405 nm), but no or minimal photoresponses when applying 2P stimulation (780 nm). On the contrary, slices incubated with **MAG**$_{2P\_F}^{slow}$ did not only show clear photoresponses upon illumination at 405 nm, but comparable light-induced signals were also recorded for the same cells by excitation at 780 nm, thus indicating very similar 1P and 2P photoswitching efficacies (Fig. 6c, e). In all the cases, the photoresponses observed upon stimulation at 405 and/or 780 nm were completely inhibited under green light illumination (514 nm), which demonstrated the long $\tau_{cis}$ of both **MAG** and **MAG**$_{2P\_F}^{slow}$ switches (see also Supplementary Movies 3-4).

To compare the efficacy of LiGluR photocontrol in slices conjugated with **MAG** or **MAG**$_{2P\_F}^{slow}$, we determined both the calcium-induced fluorescence enhancement responses after 1P and 2P stimulation (Fig. 6f) and the 2P/1P response ratio (Fig. 6g). Clearly, **MAG** could efficiently activate LiGluR and thus induce neuronal activity under 1P excitation at 405 nm, but not upon 2P stimulation at 780 nm. In contrast, **MAG**$_{2P\_F}^{slow}$ stimulation reliably activated neurons using both 1P and 2P excitation with violet and NIR light, respectively, and similar 1P and 2P signals were indeed measured at our experimental conditions. In addition, the 2P-induced control of neuronal activity with **MAG**$_{2P\_F}^{slow}$ was found to be robust and reproducible in many different cells ($n = 6$ cells) located at distinct depths of the brain tissue (0–100 μm) from hippocampal slices cultured for different days (8–15 days in vitro). Importantly, cells expressing RCaMP2 but not LiGluR-eGFP did not respond to **MAG** or **MAG**$_{2P\_F}^{slow}$ photostimulation (Supplementary Fig. 14).

**2P stimulation in vivo in *Caenorhabditis elegans*.** We further tested the ability of **MAG**$_{2P\_F}^{slow}$ to control neuronal activity in vivo using 2P excitation. For that purpose, we used *Caenorhabditis elegans* as a model of choice to analyze neuronal circuits. The morphology and function of its 302 neurons are characterized in detail, and allow scrutinizing sensory and motor circuits, among others. Figure 7 summarizes the results of all-optical experiments with GluK2-L439C-mCherry and GCaMP6s fluorescent calcium reporter co-expressed in touch receptor neurons (TRNs). From the six TRNs, which tile the receptive field of the animal in anterior/posterior and left/right, we specifically focused on the single pair PLML/R, which is located near the tail of the nematode (Fig. 7a). Like in other neurons within the nervous system, pairs of TRNs can be selectively stimulated in the anterior and posterior part of the animal using spatial light patterning, but differential unilateral activation is difficult, which poses a hurdle to photomanipulate their activity at high resolution using optogenetics[47]. This is especially relevant when neurons are overlapping along the optical axis, as shown in Fig. 7b (compare confocal projection top view, and side view of the confocal sections) or are closely packed as in the head. In such cases, 2P excitation provides a unique advantage over 1P excitation to activate cells with 10 μm axial plane selectivity[48].

Upon expression of GluK2-L439C-mCherry in TRNs, we observed localization of mCherry on the plasma membrane and in vesicles that are transported along the sensory neurite (Supplementary Movie 5). After delivery of **MAG**$_{2P\_F}^{slow}$ to living animals (see details in the Methods section), calcium activity was monitored in the posterior TRNs (PLM) expressing GluK2-L439C-mCherry and GCaMP6s using a confocal fluorescence microscope. Clear photoresponses to 2P excitation were observed in every animal responding to 1P excitation ($n = 5$ neurons from four different individuals, a 2P/1P excitation efficacy of 100%,

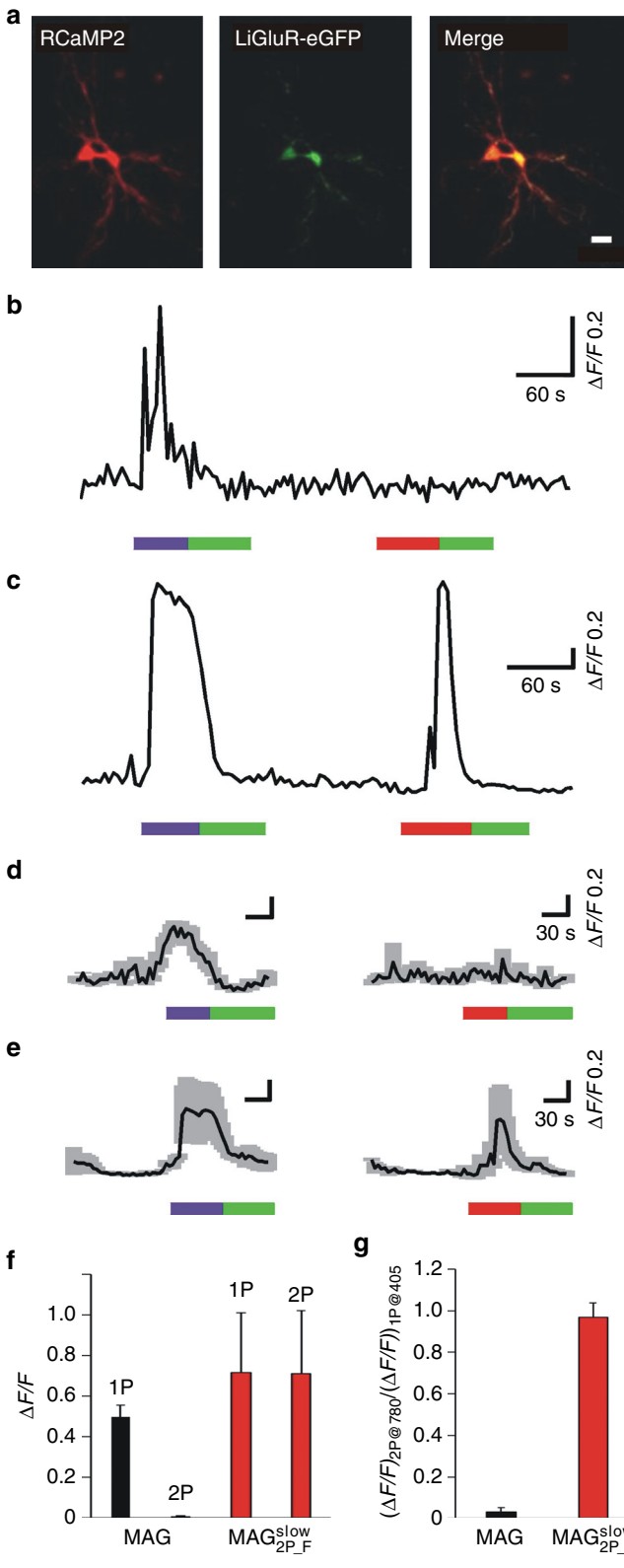

**Fig. 6** Two-photon (2P) $Ca^{2+}$ photoresponses in rat hippocampal organotypic slices expressing GluK2-L439C-eGFP and RCaMP2. **a** Microphotograph of a neuron expressing both RCaMP2 (red) and GluK2-L439C-eGFP (green) (scale bar = 20 μm). **b, c** Real time traces of a single-cell neuronal activity of slices incubated with **b MAG** or **c** $MAG^{slow}_{2P\_F}$. **d, e** Average one-photon (1P) and 2P responses of neurons incubated with **d MAG** ($n = 3$ biologically independent cells) or **e** $MAG^{slow}_{2P\_F}$ ($n = 6$ biologically independent cells). In **b–e** 1P stimulation was performed at 405 nm (purple bar, 0.81 mW μm$^{-2}$) and 514 nm (green bar, 0.35 mW μm$^{-2}$), and 2P stimulation at 780 nm (red bar, 2.8 mW μm$^{-2}$). **f, g** Quantification of photoresponses in slices incubated with **MAG** (black bars, $n = 5$ biologically independent cells) and $MAG^{slow}_{2P\_F}$ (red bars, $n = 6$ biologically independent cells): **f** fluorescence enhancement; **g** ratio between the 2P and 1P responses of **MAG** and $MAG^{slow}_{2P\_F}$ for the same cells. Error bars are SEM. Source data for **d–g** are provided as a source Data file

activity of single, individual neurons with high efficacy and selectivity.

## Discussion

Genetically targeted and pharmacologically selective[49] synthetic photoswitches offer great potential to dissect neuronal circuits based respectively on specific promoters and endogenous receptors. However, the efficiency of azobenzene-based photoswitchable ligands is ultimately limited by the *cis* isomer population that can be achieved upon irradiation, which is determined by different experimental (excitation intensity and wavelength) and photochemical parameters (*trans* and *cis* absorptivities and isomerization quantum yields, *cis* state thermal lifetime). Among them, the 2P activity of long $\tau_{cis}$ azobenzene switches is mainly controlled by the $\sigma_2$ values for both isomers at the excitation conditions, since they normally show relatively similar isomerization quantum yields (e.g. $\Phi_{trans \to cis} = 0.18$ and $\Phi_{cis \to trans} = 0.30$ for **MAG** in 99% PBS:1% DMSO, Supplementary Table 3). In the case of **MAG** and $MAG_0$ under NIR illumination, $\sigma_{2,trans}$ is low and, worse still, even smaller than $\sigma_{2,cis}$, which strongly disfavors 2P-induced *trans → cis* isomerization[34]. This leads to photostationary states with very low *cis* content and, as such, poor 2P LiGluR responses with NIR light even at high excitation intensities (~20 mW μm$^{-2}$)[33,34]. As discussed above, compounds $MAG^{slow}_{2P}$ and, especially, $MAG^{slow}_{2P\_F}$ overcome this drawback by introducing rationally designed azobenzene cores with both large $\sigma_{2,trans}$ and $\tau_{cis}$ values, which allow for robust and reliable 2P signals at milder, more cell-compatible excitation conditions (2.8 mW μm$^{-2}$) that are comparable to 1P responses.

As for $MAG_{2P}$ and $MAG_{460}$ biological photoactivity, it is mainly governed by a different factor: the millisecond *cis* state lifetime of both switches in physiological media[33,34]. Although this enables single-wavelength, fast neuronal stimulation, it dramatically limits the extent of the photostationary state reached even when using high excitation intensities. As a result, $MAG_{2P}$ and $MAG_{460}$ also lead to smaller 2P responses with NIR light than under 1P stimulation (~10–40%) despite their enhanced $\sigma_{2,trans}$ values with respect to **MAG** and $MAG_0$[33,34]. This, together with their low *cis* state lifetimes, is a severe constraint for the 2P stimulation of calcium-evoked photoresponses with $MAG_{2P}$ and $MAG_{460}$. On the contrary, the higher stability of $MAG^{slow}_{2P}$ and $MAG^{slow}_{2P\_F}$ *cis* isomers enables sustained receptor activation, which combined with their optimized 2P excitability results in larger photoresponse amplitudes and allows the manipulation of calcium-regulated processes with NIR light. Thus, while very small R-GECO1 calcium responses were described for LiGluR-expressing HeLa cells under 2P stimulation of $MAG_{460}$ ($\Delta F/F < 0.1$)[34], we obtained herein calcium-induced

Figs. 7c, e, Supplementary Movies 6–7), whereas control-treated animals (vehicle) only showed a partial fluorescence reduction due to bleaching of mCherry and GCaMP6s ($n = 5$ neurons from four different individuals, Fig. 7d). No signs of toxicity were observed after compound injection, during recovery and imaging. Thus, the rationally designed 2P excitation properties of $MAG^{slow}_{2P\_F}$ can also be used in vivo to photomanipulate neuronal

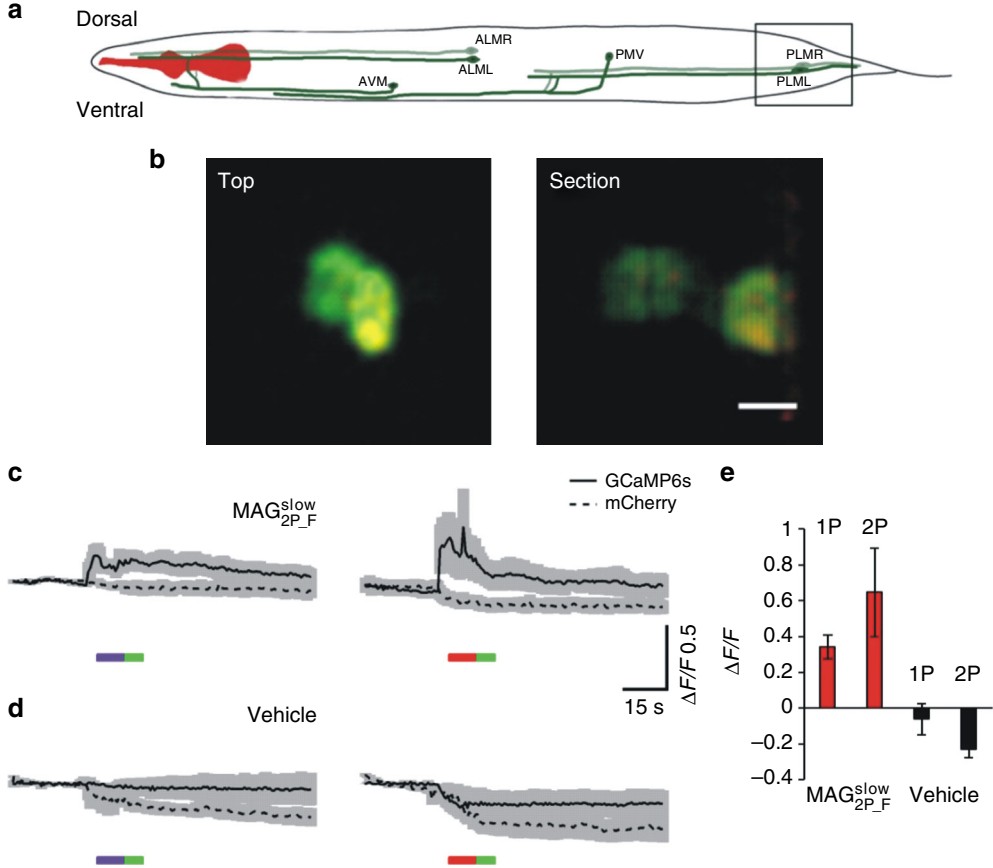

**Fig. 7** In vivo calcium induced photoresponses by two-photon (2P) stimulation of **MAG$_{2P\_F}^{slow}$** in *C. elegans*. **a** Schematics of *C. elegans* in which touch receptor neurons (TRNs) are depicted. Squared region is magnified in **b**. **b** Microphotograph of an animal expressing LiGluR-mCherry (red) and GCaMP6s (green) (scale bar = 5 μm). Top and lateral section view of TRN from the tail. **c**, **d** Average traces of one-photon (1P)- and 2P-induced photoactivation of TRNs in animals treated with **c** **MAG$_{2P\_F}^{slow}$** (*n* = 5 and 6 cells from four different animals experiments for 1P and 2P traces, respectively) or **d** with vehicle (*n* = 5 and 6 cells from four different animals experiments for 1P and 2P traces, respectively). Continuous line trace indicates GCaMP6s fluorescence signal and dashed trace mCherry fluorescence. In **c** and **d** 2P stimulation was performed at 780 nm (red bar, 2.8 mW mm$^{-2}$) and 1P stimulation at 405 nm (purple bar, 15 μW mm$^{-2}$) and 514 nm (green bar, 1.21 μW mm$^{-2}$). **e** Quantification of photoresponses (fluorescence enhancement) in animals injected with **MAG$_{2P\_F}^{slow}$** (red bars, *n* = 5 cells from four different animals experiments) and vehicle (black bars, *n* = 5 cells from four different animals experiments). Error bars are SEM. Source data for **c–e** are provided as a source Data file

fluorescence enhancement values higher than 1 for the same indicator in HEK cells upon excitation of **MAG$_{2P\_F}^{slow}$**-tethered LiGluR with NIR light.

Other advantages derive from the use of **MAG$_{2P}^{slow}$** and **MAG$_{2P\_F}^{slow}$** over **MAG$_{2P}$** and **MAG$_{460}$** for the 2P stimulation of LiGluR. Technically, *cis* isomer stability allows maintaining the glutamate moiety bound to the receptor in the dark, which best mimics the presence of a high neurotransmitter concentration in extracellular medium during presynaptic release. This approach is in contrast to the use of Concanavalin A to block receptor desensitization thus keeping the channel open (**MAG$_{2P}$**)[33], and to mutation K456A used in GluK2 to quicken receptor recovery from desensitization[34], which results in higher and longer-lasting currents during illumination of a *cis*-unstable photoswitch (**MAG$_{460}$**). Such sustained calcium influx facilitates photoresponse detection in imaging experiments but could be toxic for the neurons, and is not physiological. In our case, we used Concanavalin A in order to quantify steady-state currents in recordings with cell lines (Figs. 3–5), and to obtain clear responses in the first in vivo studies (Fig. 7), but physiological desensitization was always preserved in all the experiments with rat brain slices (Fig. 6). Likely, the *cis* isomers of **MAG$_{2P}^{slow}$** and

**MAG$_{2P\_F}^{slow}$** cause receptor photoactivation, channel opening, and closing by desensitization, which limits the cytotoxic effects of calcium influx. The stability of *cis* isomers also allows shortening illumination pulses and reducing phototoxicity. The repetitive photoresponses that we obtained in slices and in vivo are in agreement with the low toxicity of our compounds. These robust calcium responses mediated by desensitizing receptors could be related to triggering of intracellular processes[50] and/or changes in synaptic receptor mobility[51]. These processes are involved in neuronal plasticity and are currently under investigation.

Photocontrol of endogenous neuronal receptors[52] and intracellular presynaptic proteins[53] has been recently shown in *C. elegans* using 1P excitation. Our 2P excitation experiments in vivo with **MAG$_{2P\_F}^{slow}$** conjugated to GluK2-L439C take advantage of genetic manipulation for demonstration purposes (including the expression of fluorescent calcium reporters), but the robust photoresponses obtained (Fig. 7, Supplementary Movies 6-7) suggest that introducing our *ortho*-fluorosubstituted azobenzene cores in freely diffusible photoswitches would also allow photocontrolling endogenous neuronal receptors and signaling proteins with high efficacy and spatial confinement. 2P excitation is currently the only method that allows photostimulation with axial

plane selectivity at micrometer resolution[48] and thus has an advantage over 1P excitation to activate individual neurons in clustered 3D structures like ganglia and brain tissue. Thus, photoswitches that combine high 2P excitation efficacy and pharmacological selectivity will be an invaluable tool to investigate intact neuronal circuits and subcellular signaling pathways, and a powerful complement to optogenetic manipulation techniques[54].

Thus, we have shown that 2P-optimized azobenzene photoswitches are an important complement to 2P-enabled caged ligands like MNI-glutamate[49] or optogenetic tools like C1V1[48] and ChR2-H134R[55], but they have advantages of their own. Tethered MAG derivatives allow reversible activation of glutamate receptors without the need of perfusing high concentrations of caged glutamate compounds and the disadvantages of glutamate and byproduct spillover upon uncaging. Moreover, glutamate receptors have larger cation conductances than channelrhodopsins[56], do not require illuminating large cellular regions[57], and can be targeted by MAG-like photoswitches at their physiological location[58]. Finally, the 2P-optimized azobenzene core reported here entails minimal structural modifications that can be grafted into other azobenzene-based photoswitches, provided that their pharmacological properties are not altered, and thus these findings have direct and general application to light-regulated ligands[2–4].

In conclusion, we rationally designed azobenzene chromophores based on theoretical calculations to present both high 2P absorptivity of NIR light and long *cis* state lifetime, which were then exploited in the synthesis of PTLs $\mathbf{MAG_{2P}^{slow}}$ and $\mathbf{MAG_{2P\_F}^{slow}}$. Optimized 2P stimulation of light-gated ionotropic glutamate receptors was accomplished with these compounds, which far surpassed the performance of other azobenzene-based photoswitches previously assayed for the manipulation of neuronal tissues under multiphoton excitation conditions. Taking advantage of the high 2P excitability of $\mathbf{MAG_{2P}^{slow}}$ and $\mathbf{MAG_{2P\_F}^{slow}}$ with NIR radiation, reliable and sustained photocontrol over the activity of neurons could be attained in light-scattering tissue both in vitro and in vivo, such as in brain slices under nearly physiological conditions and *C. elegans* TRNs. The results presented here constitute a proof of concept that paves the way toward all-optical experiments of neuronal activity imaging and manipulation in vivo using azobenzene photoswitches, which ultimately require the use of multiphoton excitation with NIR light[59].

## Methods

**Theoretical calculations.** All calculations were carried out at the DFT (for ground electronic states) and TDDFT (for excited electronic states) levels using the long-range corrected hybrid CAM-B3LYP functional, which is known to correctly describe excited states of charge-transfer type[60]. 6–31G(d), a split-valence basis set with polarization functions in heavy atoms, was used in all the cases. Twenty excited electronic states of the same multiplicity as the ground electronic state (singlet) were converged in the TDDFT calculations. Solvent (water) effects were introduced through the self-consistent polarizable continuum model continuum method[61]. To compute 2P absorption properties, the density functional response theory was employed to calculate the 2P transition matrix elements. 2P absorption cross sections were then estimated through the following expression[62]:

$$\sigma_2(\omega) = \frac{8\pi^2 \alpha a_0^5 \omega^2}{c\Gamma} \delta(\omega) \qquad (1)$$

where $\alpha$ is the fine structure constant, $a_0$ the Bohr radius, $c$ is the speed of light in vacuum, $\Gamma$ is the full width at half maximum of the Lorentzian line-shape broadening, and $\delta(\omega)$ is the 2P absorption transition probability calculated through the response theory assuming linearly polarized excitation light. We set $\Gamma = 0.2$ eV, which reasonably agrees with the value experimentally determined for *trans*-$\mathbf{MAG_{460}}$ (~0.15 eV[34]). Here we also considered the most common case of a degenerate 2P absorption process where $\omega$ is half of the transition frequency of the excited state. All TDDFT calculations and evaluation of 2P absorption properties were calculated using the Dalton suite of programs[63]. Optimizations and evaluation of energy barriers in the ground electronic state were carried out with the

GAUSSIAN09 program[64]. For energy barrier calculations of thermal *cis-trans* isomerization, we investigated both the rotation and inversion mechanisms. In the case of asymmetric azobenzenes, the two feasible paths leading to inversion were analyzed.

**Synthesis.** A detailed description of the synthesis of $\mathbf{MAG_{2P}^{slow}}$ and $\mathbf{MAG_{2P\_F}^{slow}}$ is given in the Supplementary Methods.

**Photochemical characterization.** *Trans-cis* isomerization of $\mathbf{MAG_{2P}^{slow}}$, $\mathbf{MAG_{2P\_F}^{slow}}$, and their separated azobenzene photochromes in solution was investigated by: (i) $^1$H NMR for the elucidation of the photostationary state mixtures in organic solvents; and (ii) steady-state UV-vis absorption spectroscopy for *trans-cis* photoisomerization in aqueous media and slow *cis-trans* thermal back-isomerization processes.

**Cell culture.** Tsa201 cells were purchased from the European Collection of Authenticated Cell Culture. HEK293 tsA201 cells were plated on glass coverslips and transfected with GluK2-L439C-eGFP or co-transfected with GluK2-L439C and GCaMP6s or R-GECO1. Prior to each experiment, coverslips were incubated with the PTL of choice to allow chemical conjugation to the receptor channel. A second incubation with Concavalin A was done to inhibit desensitization of the glutamate receptor. Cells were mounted on the recording chamber filled with a bath solution composed of (in mM): 140 NaCl, 1 MgCl$_2$, 2.5 KCl, 10 4-(2-hydroxyethyl)-1-piperazineethanesulfonic acid (HEPES), 2.5 CaCl$_2$, and 10–20 glucose to fix osmolarity to 310 mOsm kg$^{-1}$. NaOH was added to adjust the pH to 7.42. To activate GluK2-L439C, 300 µM glutamate in bath solution was perfused.

**Electrophysiology.** Voltage-clamp recordings under whole-cell configuration were acquired at 1 kHz. Borosilicate glass pipettes were pulled with a typical resistance of 4–6 MOhm and filled with a solution containing (in mM): 120 cesium methanesulfonate, 10 tetraethylamonium chloride, 5 MgCl$_2$, 3 Na$_2$ATP, 1 Na$_3$GTP, 20 HEPES, 0.5 EGTA; osmolarity was 290 mOsm kg$^{-1}$ and pH 7.2 was adjusted with CsOH. Cell membrane voltage was held at −70 mV. Photostimulation during electrophysiological recordings was induced by illuminating the entire focused field with monochromatic light in an inverted microscope (power density: 22.0 µW mm$^{-2}$ at 380 nm, 45.9 µW mm$^{-2}$ at 460 nm, and 47.4 µW mm$^{-2}$ at 500 nm).

**Calcium imaging and 1P stimulation.** Cells were imaged on an inverted fully motorized digital microscope at room temperature with a frame rate of 2 s and exciting GCaMP6s at 490 nm during 10 ms. Photoisomerization was achieved by illuminating the focused sample with flashes of violet (380 nm, 0.5 s duration) and green (500 nm, 0.5 s duration) light for activation and deactivation, respectively. Calcium imaging activation spectra under 1P stimulation ranged from 280 to 480 nm at 20 nm steps. Light flashes were nested in between GCaMP6s fluorescence measurements. Photostimulation intervals lasted a total of 3.2 min for activation and 2.4 min for deactivation. At the end of this protocol, 300 µM free glutamate solution was added and the calcium imaging response was measured.

**Calcium imaging and 2P stimulation in HEK cells.** 2P experiments were performed in the Advanced Digital Microscopy Core Facility of IRB Barcelona with a confocal multiphoton microscope equipped with a 80 MHz Ti:Sapphire for 2P stimulation with NIR light (710–990 nm), and cw diode (405 nm) and Ar (514 nm) lasers for 1P stimulation and calcium imaging with visible light. In these experiments, R-GECO1 was used as a Ca$^{2+}$ fluorescent indicator instead of GCaMP6s because it does not absorb at 405 nm, the excitation wavelength used to test the 1P activity of LiGluR. Imaging of R-GECO1 was done at 514 nm with a frame rate of 4 s. Photostimulation was achieved by raster-scanning the tightly focused laser of choice over a selected area of the field of view: 405 nm (0.37 mW µm$^{-2}$) for 1P activation, 514 nm (0.35 mW µm$^{-2}$) for 1P deactivation, and 720–840 nm (2.8 mW µm$^{-2}$) for 2P activation of LiGluR. Photostimulation scans were fit to keep imaging interval, and illumination periods at a given wavelength lasted in total 1 min. Interstimulus imaging periods also lasted 1 min.

**Calcium imaging and 2P stimulation in hippocampal slices.** All procedures were conducted in accordance with the European Guidelines for Animal Care and Use in Research and were approved by the Animal Experimentation Ethics Committee at the University of Barcelona (Spain). Hippocampal organotypic slices of 400 µm in thickness were obtained from postnatal day 6–8 rats and cultured for 5–7 days until biolistically transfected with RCaMP2a and GluK2-L439C-eGFP, as described in ref. [46]. Although transfection with R-GECO1 was also assayed, RCaMP2a was finally used as a fluorescent indicator in hippocampal slice cultures because of the higher level of expression achieved. Before each experiment, slices were incubated with the PTL of choice for 10 min in artificial cerebrospinal fluid (ACSF) containing 119 mM NaCl, 2.5 mM KCl, 3 mM CaCl$_2$, 0.2 mM MgCl$_2$, 26.2 mM NaHCO$_3$, 1 mM NaH$_2$PO$_4$, and 11 mM glucose, equilibrated with 5% CO$_2$/95% O$_2$. After washes with fresh ACSF, slices were placed on the recording chamber and were continuously perfused with ACSF. RCaMP2a was excited at 561 nm with an

imaging interval of 4 s. eGFP was excited at 488 nm. Photostimulation was achieved by raster-scanning the tightly focused laser of choice over the whole field of view at 400 Hz: 405 nm (0.81 mW $\mu m^{-2}$) for 1P activation, 514 nm (0.35 mW $\mu m^{-2}$) for 1P deactivation, and 780 nm (2.8 mW $\mu m^{-2}$) for 2P activation of LiGluR. Photostimulation scans were fit to keep imaging interval, and illumination periods at a given wavelength lasted in total 1 min. Interstimulus imaging periods lasted 1.5 min.

**Calcium imaging and 2P stimulation in vivo in C. elegans.** Animal experiments carried out have been approved by the ethics committee of the ERC and the animal welfare has been respected. C. elegans are nematodes and therefore not covered by EU Directive 2010/63/EU, which only covers vertebrates and cephalopods. Likewise, worms are not covered by the Spanish "Código de protección y bienestar animal". For these reasons, there is no need for authorizations, personal licenses, standards for procedures or detailed description of number of animals to be used, nature of experiments or anticipated impact, and minimization thereof. We generated strain MSB104 [mirEx22(mec-17p::iGluR6::mCherry;myo-2p::mCherry); ljSi123(mec-7p:GCaMP6s::SL2::tagRFP);lite-1(ce314)X]. Transgenesis was performed according to standard methods for microinjection[65] by microinjecting a DNA mix containing 50 ng $\mu l^{-1}$ pNMSB18 (mec-17p::iGluR6::mCherry, Supplementary Table 7), 1.5 ng $\mu l^{-1}$ myo-2p:mCherry as a coinjection marker, and 50 ng $\mu l^{-1}$ 1Kb Plus DNA ladder (Invitrogen) as a carrier into the gonad of GN692 young adult worms[66]. $\text{MAG}_{2P\_F}^{slow}$ (10 mM in M9 buffer and 0.3 mg ml$^{-1}$ ConA) was administered to the animals by microinjection into the body cavity. Control group was performed microinjecting with vehicle (10% DMSO and 0.3 mg/ml ConA in M9 buffer), and allowed to recover. Only roaming worms surviving the treatment were considered for the following imaging experiments. After 4 h of compound administration, TRN neurons co-expressing GluK2-L439C-mCherry and GCaMP6s were imaged in a single focused plane. Calcium imaging was performed on a Leica confocal microscope (SP5) through a ×63/1.4-numerical aperture oil objective (HCX PL APO, Leica). Calcium-sensitive GCaMP6s and calcium-insensitive red fluorescent proteins (RFPs) were simultaneously excited at 488 and 561 nm for 343 ms, using bidirectional laser scanning at 400 Hz. Images were recorded with a resolution of 512 × 512 and a digital zoom of 4, with an imaging interval of 660 ms. GCaMP6s and RFP fluorescence were recorded with two different HyD detectors with a detection range from 500 to 550 nm and from 569 to 648 nm, respectively. Pinhole aperture was set at ~500 μm. Whole-field photostimulation flashes were fit to keep imaging interval. Photostimulation was done at 256 × 256 resolution with bidirectional laser scan, with a digital zoom of 4. One-photon photostimulation was done at 405 nm (15 μW $\mu m^{-2}$), and 2P stimulation at 780 nm (2.8 mW $\mu m^{-2}$). Back-photoisomerization was achieved at 514 nm (1.21 μW $\mu m^{-2}$). Intensity and duration of the photostimulation intervals were adjusted to obtain the optimal photoresponse and reproducibility.

**Reporting summary.** Further information on experimental design is available in the Nature Research Reporting Summary linked to this article.

## Data availability

The authors declare that the main data supporting the findings of this study are available within the article and its Supplementary Information. Extra data are available from the corresponding author upon request.

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

## Acknowledgements

G.C. acknowledges the "Generalitat de Catalunya" for her pre-doctoral FI grant. A.G.-C. was supported by fellowship BES-2014–068169. M.B. was supported by a Marie Curie Reintegration Grant (H2020-MSCA-IF). We are grateful to Ehud Isacoff (UC Berkeley) for sharing the LiGluR clone, to Dirk Trauner (NYU) for providing **MAG** compound, and to Anna Lledó i Lidia Bardia (IRBB imaging facility) for support in imaging experiments. We would like to acknowledge César Alonso, Angel Sandoval, and Merche Rivas from the Biolab at ICFO for technical support in C. elegans experiments. We also acknowledge financial support from AGAUR/Generalitat de Catalunya (CERCA Programme and projects 2017-SGR-00465, 2017-SGR-1442, and 2017-SGR-1012), Severo Ochoa (SEV-2015–0522), Fundacion Privada Cellex, FEDER funds, ERANET SynBio MODULIGHTOR and Human Brain Project WAVESCALES projects, MINECO/FEDER (projects CTQ2015–65439-R, CTQ2016-80066-R, CTQ2016- 75363-R, CTQ2017-83745-P, and RYC-2015-17935), and Fundaluce and Ramón Areces foundations. M.K. acknowledges support through HFSP CDA and ERC MechanoSystems (715243).

## Author contributions

P.G., J.H., F.B. and R.A. conceived and supervised the project. J.M.L., M.M. and R.G. performed the theoretical calculations. R.A., F.B. and J.H. designed and supervised the chemical synthesis. J.H. designed and supervised the photochemical study. G.C. performed the synthesis and the photochemical study. M.G.-M. collaborated in the synthesis. A.G.-C. and N.C. prepared cultured HEK cells. A.G.-C. and G.C. performed the study on cultured HEK cells. A.G.-C. and M.B. performed the experiments with neurons. M.K. and M.P.R designed and supervised the experiments in vivo with C. elegans nematodes. A.G.-C. and M.P.R performed the experiments in vivo with C. elegans worms. All the authors contributed to the preparation of the article.

## Additional information

**Competing interest:** The authors declare no competing interest.

