## [Peer Review File · Nature Communications]

Reviewers' comments:

Reviewer #1 (Remarks to the Author):

The paper entitled «Rational designed azobenzene photoswitches for reliable two-photon neuronal excitation» by P. Gorostiza, R. Alibés et coll. reports the rational design, the synthesis and the characterization of the photophysical properties of azobenzene photoswitches together with their uses on neuronal tissues. The authors present new photoswitches with high two-photon sensitivity and enhanced cis isomer thermal lifetime. This unique behavior allows for the first time to monitor one of the physiological effect (Ca^{2+} concentration increase) of the local excitation of GluK2 receptors on hippocampal organotypic slices.

Therefore, this original behavior represents an important complement to 2P sensitive optopharmacologic or optogenetic tools.

The manuscript is well written. The synthesis, the TDDFT calculations and the photophysical characterizations of those new MAG derivatives together with their uses to dissect intact neuronal activities are extremely well-described and discussed. The work is original and carried out to a high standard. However, there are some corrections that should be addressed before publication. Issues to address in order to elevate the rigor and impact of the study are listed below.

1) The slow cis isomer thermal lifetime of the MAGslow compounds should be studied not only in a cuvette by also using tethered MAGs compounds on GluK2-L439C receptors. Therefore in my opinion electrophysiological traces showing the 1P thermal relaxation of respectively MAG2Pslow, MAG2P-Fslow and MAG2P are missing.

2) Since MAG460 compound (in ref 45) shows similar 2P excitability compare to the new MAGs described in this manuscript, it's difficult to evaluate the enhanced 2P excitability of those compounds on GluK2 receptors. In my opinion, in fig 4 the 2P activity using MAG2P should also be presented and discussed. Otherwise, the authors should replace in the text "enhanced 2P excitability" by "high 2P excitability".

3) Technically, I agree that the cis-isomer stability is important to study the neuronal plasticity with MAGs using 2P excitation. But the author should also discuss on the physiological influence of "long time" GluK2 channels opening.

4) In the SI the quality of the 2D NMR figures should be improved.

In Summary, I think there is great potential here. In my opinion, this manuscript is highly significant to deserve a publication in this journal after the authors address some corrections.

Reviewer #2 (Remarks to the Author):

This manuscript by Cabre et al makes a valuable contribution to the field of photo-control of biological systems. However, major revision of the text, and further computational work will be required to make it a convincing story of rational design.

The authors argue that effective 2P photoswitching requires a good 2P cross section for trans-to-cis switching, and a slow enough thermal cis-to-trans half life so that substantial production of the cis isomer is possible. They argue that a compromise must be met – not too much of a push pull system of the thermal rate will be too high, but enough of a push-pull system so that the asymmetry produces a practical 2P cross-section. This makes sense, but, as the authors mention in the discussion, the cis 2P cross section and wavelength dependence also matter. This should be introduced right at the start of the rational design process. Were 2P cross sections calculated for the cis isomers of the new structures? These should be reported. I suspect the accuracies of these are not great, so that ultimately experiments are required, but it is disingenuous to claim this system is entirely predictable. Also no mention is made of why the single ortho F substituent was chosen. The authors state (line 86- our approach towards this goal relied on the accurate selection of the substitution patten of the azoaromatic core of the system), but only 1 (single ortho) substitution patterns were calculated. A real computational design process would have explored

several variations 1,2,3,4 F in various positions (as has been done by Bleger & Hecht). In some cases these have dramatic effects on thermal cis-to-trans barriers. Also, a much more detailed description of the barrier high calculation is required. What mechanisms of isomerization were explored (inversion, rotation, a mix?).

Smaller points:

It is not clear what is meant by this statement:

“Technically, cis-isomer stability facilitates raster scanning laser stimulation without requiring super-fast spinning disk microscopes. In addition, MAG2Pslow and MAG2P_Fslow display remarkable photostability, providing sustained photoresponses over time that can be useful to study neuronal plasticity phenomena such as long-term potentiation and depression.

Claims of “100% reliability” (e.g. line 24) sound overstated and meaningless. Even the word reliable in the title should be removed – it is unclear what it means.

Reviewer #3 (Remarks to the Author):

This paper reports the development of a highly 2P-sensitive azobenzene photoswitch for control of glutamate receptors. The new compound has a higher 2P absorption cross-section than previously synthesized molecules. The invention of 2-photon fluorescent probes for calcium and many biochemical processes have been a boon not only for neurobiology but cell biology and other fields. 2P calcium indicators have become very important because they generate signals that indirectly reflect action potential firing in neurons deep in the brain.

The reasoning here is that 2P-sensitive photoswitches could also be important for allowing control of synaptic receptors deeper and more precisely than with previous tools. The paper makes this conceptual argument, but unfortunately, provides almost no functional evidence. For this paper to have a significant impact, it would need to convincingly demonstrate that 2-photon control can be exerted in vivo, where the advantages of 2P might really make a difference. More to the point, the paper would need to answer an important question that is currently inaccessible by other means.

In addition to the system application, very little is said about how the photoswitch actually works on glutamate receptors, or whether it alters the effects of the natural neurotransmitter glutamate on the receptor. The only functional measurements in the paper come from calcium imaging, a very indirect way to study and understand neurotransmitter receptors. Fundamental properties of the receptors, including apparent affinity for the probe (and for glutamate), activation and deactivation kinetics, and the extent and kinetics of desensitization, are not obtainable with optical imaging, and are not addressed in the paper. As it stands, the paper is very incomplete, and there is more work to do before a compelling case could be made for the new tool having a significant impact.

SUMMARY OF CHANGES

a) Reviewer 1

The paper entitled «Rational designed azobenzene photoswitches for reliable two-photon neuronal excitation» by P. Gorostiza, R. Alibés et coll. reports the rational design, the synthesis and the characterization of the photophysical properties of azobenzene photoswitches together with their uses on neuronal tissues. The authors present new photoswitches with high two-photon sensitivity and enhanced cis isomer thermal lifetime. This unique behavior allows for the first time to monitor one of the physiological effect (Ca²⁺ concentration increase) of the local excitation of GluK2 receptors on hippocampal organotypic slices.

Therefore, this original behavior represents an important complement to 2P sensitive optopharmacologic or optogenetic tools.

The manuscript is well written. The synthesis, the TDDFT calculations and the photophysical characterizations of those new MAG derivatives together with their uses to dissect intact neuronal activities are extremely well-described and discussed. The work is original and carried out to a high standard.

We appreciate the reviewer's comments and we have made the requested changes in the manuscript (highlighted). We reply to the specific questions below.

However, there are some corrections that should be addressed before publication. Issues to address in order to elevate the rigor and impact of the study are listed below.

1) The slow cis isomer thermal lifetime of the MAG_{slow} compounds should be studied not only in a cuvette by also using tethered MAGs compounds on GluK2-L439C receptors. Therefore in my opinion electrophysiological traces showing the 1P thermal relaxation of respectively MAG_{2Pslow}, MAG_{2P-Fslow} and MAG_{2P} are missing.

We agree that for biological relevance, the thermal relaxation lifetime of the photoswitches should be measured after conjugation to the receptor protein. We have thus studied the time course of photoswitch relaxation using patch clamp recordings in HEK293 cells expressing GluK2-L439C. We compared MAG (Gorostiza et al, PNAS 2007, Ref: 37), MAG_{2P^{slow}} and MAG_{2P_F^{slow}} (the thermal relaxation lifetimes of which in cuvette are approximately 10 min at room temperature, Fig. S4) to MAG_{2P} (Izquierdo-Serra et al, JACS 2014, Ref: 45). The results are shown in the new Supplementary Fig. S6 and briefly commented in the main text (1st paragraph, page 12).

Current recordings of MAG_{2P^{slow}}, MAG_{2P_F^{slow}} and MAG during 10 min periods in the dark and at room temperature, and the corresponding exponential fits (indicated by grey curves in Supplementary Fig. S6a-c, respectively) yield lifetimes in full agreement with the measurements in cuvette and with reported results for MAG (Gorostiza et al., PNAS 2007, Ref: 37). In particular, we obtained $\tau = 570 \pm 14$ s (MAG_{2P^{slow}}), 534 ± 3 s (MAG_{2P_F^{slow}}), and 900 ± 54 s (MAG). In contrast, fast-relaxing photoswitch MAG_{2P} yield an average lifetime $\tau = 0.25 \pm 0.013$ s in the dark (Fig. S6d), also in agreement with previous reports (Izquierdo-Serra et al, JACS 2014, Ref: 45).

Note that long-lasting current recordings (Fig. S6a-c) display partial current run-down, as shown by the 380 nm light pulse at the end of the current trace. This is usually due to cell dialysis in whole cell patch clamp configuration, but in our case it does not cause overestimation of lifetimes. Nevertheless, in order to reduce rundown and improve gigaseal stability, we also studied the photoswitch thermal relaxation in shorter periods (2 min in the dark, Fig. S6e-f). Although these recordings do not allow calculating reliable lifetime values, they clearly show that MAG_{2P^{slow}} and MAG_{2P_F^{slow}} induce stable channel activation with short light pulses, and do not require

continuous illumination as $\text{MAG}_{2\text{P}}$ or MAG_{460} . For this reason, $\text{MAG}_{2\text{P}}^{\text{slow}}$ and $\text{MAG}_{2\text{P}_F}^{\text{slow}}$ afford higher 2P excitation efficacy, as originally designed and demonstrated in subsequent figures.

2) Since MAG_{460} compound (in ref 45) shows similar 2P excitability compare to the new MAGs described in this manuscript, it's difficult to evaluate the enhanced 2P excitability of those compounds on GluK2 receptors. In my opinion, in fig 4 the 2P activity using $\text{MAG}_{2\text{P}}$ should also be presented and discussed. Otherwise, the authors should replace in the text "enhanced 2P excitability" by "high 2P excitability".

We agree that a direct evaluation of the enhancement of 2P excitability is difficult, due to the low calcium photoresponses of $\text{MAG}_{2\text{P}}$, and we have replaced "enhanced 2P excitability" by "high 2P excitability" throughout the text.

3) Technically, I agree that the *cis*-isomer stability is important to study the neuronal plasticity with MAGs using 2P excitation. But the author should also discuss on the physiological influence of "long time" GluK2 channels opening.

Following the reviewer's request, we have extended and improved this paragraph in the text (2nd paragraph, page 22). The idea is that the high stability of the *cis* isomers of $\text{MAG}_{2\text{P}}^{\text{slow}}$ and $\text{MAG}_{2\text{P}_F}^{\text{slow}}$ allows maintaining the glutamate moiety bound to the receptor in the dark, which best mimics the presence of a high neurotransmitter concentration in the extracellular medium during presynaptic release. This approach is in contrast to the use of Concanavalin A to block receptor desensitization thus keeping the channel open (Izquierdo-Serra et al, JACS 2014, Ref: 45), and to mutation K456A introduced in GluK2 by Carroll et al, PNAS 2015 to quicken receptor recovery from desensitization, which results in higher and longer-lasting currents during illumination of a *cis*-unstable photoswitch (MAG_{460}). Such sustained calcium influx facilitates photoresponse detection in imaging experiments but could be toxic for the neurons, and is not physiological. In our case, we used Concanavalin A in order to quantify steady-state currents in recordings with cell lines, and to obtain clear responses in the first *in vivo* studies, but physiological desensitization was always preserved in all experiments with rat brain slices. Likely, the *cis* isomers of $\text{MAG}_{2\text{P}}^{\text{slow}}$ and $\text{MAG}_{2\text{P}_F}^{\text{slow}}$ cause receptor photoactivation, channel opening, and closing by desensitization, which limits the cytotoxic effects of calcium influx. The stability of *cis* isomers also allows shortening illumination pulses and reducing phototoxicity. The repetitive photoresponses that we obtain in slices and *in vivo* are in agreement with the low toxicity of our compounds. These robust calcium responses mediated by desensitizing receptors could be related to triggering of intracellular processes (Lauri et al, Neuron 2003, Ref: 64) and/or changes in synaptic receptor mobility (de Luca et al, Neuron 2017, Ref: 65). These processes are involved in neuronal plasticity and are currently under investigation.

4) In the SI the quality of the 2D NMR figures should be improved.

As requested by the reviewer, we have improved the quality of the 2D NMR spectra in the SI.

In Summary, I think there is great potential here. In my opinion, this manuscript is highly significant to deserve a publication in this journal after the authors address some corrections.

We hope to have addressed all the reviewer's comments and we are grateful for his/her help to improve the manuscript and figures.

b) Reviewer 2

This manuscript by Cabré et al makes a valuable contribution to the field of photo-control of biological systems. However, major revision of the text, and further computational work will be required to make it a convincing story of rational design.

We are grateful for the reviewer's positive comments and we have made the requested changes in the manuscript, including new calculations and improved description of the molecular design principles and implementation.

The authors argue that effective 2P photoswitching requires a good 2P cross section for trans-to-cis switching, and a slow enough thermal cis-to-trans half life so that substantial production of the cis isomer is possible. They argue that a compromise must be met – not too much of a push pull system of the thermal rate will be too high, but enough of a push-pull system so that the asymmetry produces a practical 2P cross-section. This makes sense, but, as the authors mention in the discussion, the cis 2P cross section and wavelength dependence also matter. This should be introduced right at the start of the rational design process. Were 2P cross sections calculated for the cis isomers of the new structures? These should be reported. I suspect the accuracies of these are not great, so that ultimately experiments are required, but it is disingenuous to claim this system is entirely predictable.

As suggested by the reviewer, we now report the computed excitation energies and 2P cross-sections for the cis isomers of the azo model compounds considered (see Tables S1-S2 in the SI). Basically, we observed that the $\sigma_{2,trans}/\sigma_{2,cis}$ ratio calculated for the new azobenzene chromophores proposed in our work (Azo1-Azo2) are significantly larger than those for previous MAG/MAG₀ and MAG_{2P}/MAG₄₆₀ compounds. This, in combination with the rather large differences in excitation energy for the 2P-allowed transition of their both isomers, should enable more efficient and selective 2P excitation of trans-Azo1-Azo2. This discussion is now given in the main text of the article when describing the computational results obtained (1st paragraph, page 8).

Also no mention is made of why the single ortho F substituent was chosen. The authors state (line 86- our approach towards this goal relied on the accurate selection of the substitution patten of the azoaromatic core of the system), but only 1 (single ortho) substitution patterns were calculated. A real computational design process would have explored several variations 1,2,3,4 F in various positions (as has been done by Bleger & Hecht). In some cases these have dramatic effects on thermal cis-to-trans barriers.

In accordance with the reviewer's request, we now report the photophysical properties for two additional azo model compounds: a) Azo3, bearing 2 ortho-fluorine substituents at the electron-poor aryl ring of the azobenzene chromophore, and b) Azo5, bearing 4 ortho-fluorine substituents on both aryl groups (see Table 1 in the main text and Tables S1-S2 in the SI). As expected from previous reports from Hecht's group, introduction of 4 ortho-fluorine substituents in Azo5 dramatically increases the thermal stability of the cis isomer with respect to Azo1-Azo3. Unfortunately, it also implies electronic symmetrization of the azoaromatic core, which results in a clear decrease of $\sigma_{2,trans}$ (and of the $\sigma_{2,trans}/\sigma_{2,cis}$ ratio). Therefore, it is less suited for 2P excitation, in agreement with our molecular design principles. This does not occur for Azo3, where the introduction of an additional ortho-fluorine at the electron-poor aryl ring of the azo group results in higher electronic desymmetrization without lowering τ_{cis} . As such, a slight increase in $\sigma_{2,trans}$ (and of the $\sigma_{2,trans}/\sigma_{2,cis}$ ratio) was calculated for Azo3 with respect to Azo1-Azo2, thus validating our strategy for the rational development of 2P-responsive MAG-inspired photoswitches. In the case of Azo3, however, we also observed a smaller difference in the excitation energy for the 2P-allowed transition of its trans and cis isomers, which should hamper selective

excitation of *trans*-Azo3 with NIR light and, consequently, counteract the effects of the rise in $\sigma_{2,trans}$ with respect to Azo1-Azo2. All this discussion is now provided in the main text of the article (pages 6-8).

Also, a much more detailed description of the barrier high calculation is required. What mechanisms of isomerization were explored (inversion, rotation, a mix?).

In Table 1 and Tables S1-S2, barrier heights are given for the lowest-energy transition state regulating the thermal *cis-trans* isomerization of each of the azo model compounds considered. For this, we investigated both rotation and inversion mechanisms and, in the case of asymmetric azobenzenes, the two possible inversion mechanisms were analyzed. In all the cases, the lowest energy path for thermal *cis-trans* isomerization was found to proceed via an inversion mechanism. This is now clarified in the Table 1 and Tables S1-S2 and in the Methods section (last paragraph, page 24).

Smaller points:

It is not clear what is meant by this statement: “Technically, cis-isomer stability facilitates raster scanning laser stimulation without requiring super-fast spinning disk microscopes. In addition, MAG2P_{slow} and MAG2P_{Fslow} display remarkable photostability, providing sustained photoresponses over time that can be useful to study neuronal plasticity phenomena such as long-term potentiation and depression.”

We apologize for this confusing paragraph, which was indeed too short to be clear. We have improved and expanded the discussion of this point in the text (2nd paragraph, page 22)

Briefly, the high stability of the *cis* isomers of MAG_{2P}^{slow} and MAG_{2P_F}^{slow} upon short illumination allows maintaining the glutamate moiety bound to the receptor in the dark, which best mimics the presence of a high neurotransmitter concentration in extracellular medium during presynaptic release. This approach is in contrast to the use of concanavalin A to block receptor desensitization thus keeping the channel open (Izquierdo-Serra et al, JACS 2014, Ref: 45), and to mutation K456A introduced in GluK2 by Carroll et al (PNAS 2015, Ref: 46) to quicken receptor recovery from desensitization, which results in higher and longer-lasting currents during illumination of a *cis*-unstable photoswitch (MAG₄₆₀). Such sustained calcium influx facilitates photoresponse detection in imaging experiments but could be toxic for the neurons, and is not physiological. In our case, we used concanavalin in order to quantify steady-state currents in recordings with cell lines, and to obtain clear responses in the first *in vivo* studies, but physiological desensitization was always preserved in all experiments with rat brain slices. The stability of *cis* isomers also allows shortening illumination pulses and reducing phototoxicity.

Claims of “100% reliability” (e.g. line 24) sound overstated and meaningless. Even the word reliable in the title should be removed – it is unclear what it means.

We used this term to refer to the fact that, when using our new compounds MAG_{2P}^{slow} and MAG_{2P_F}^{slow}, 2P calcium responses could be observed in all the cells that responded to 1P stimulation, whereas for previous MAG-type molecules only a fraction of 1P-responding cells display 2P responses. To avoid confusion, we have removed the claim of “100% reliability” from the abstract and the word “reliable” from the title.

c) Reviewer 3

This paper reports the development of a highly 2P-sensitive azobenzene photoswitch for control of glutamate receptors. The new compound has a higher 2P absorption cross-section than previously synthesized molecules. The invention of 2-photon fluorescent probes for calcium and many biochemical processes have been a boon not only for neurobiology but cell biology and other fields. 2P calcium indicators have become very important because they generate signals that indirectly reflect action potential firing in neurons deep in the brain.

*The reasoning here is that 2P-sensitive photoswitches could also be important for allowing control of synaptic receptors deeper and more precisely than with previous tools. The paper makes this conceptual argument, but unfortunately, provides almost no functional evidence. For this paper to have a significant impact, it would need to convincingly demonstrate that 2-photon control can be exerted *in vivo*, where the advantages of 2P might really make a difference. More to the point, the paper would need to answer an important question that is currently inaccessible by other means.*

Although the novel contribution of this article is the rational design, chemical synthesis and biological activity of photoswitches displaying optimized 2P responses, we acknowledge that a higher significance and impact will be achieved by demonstrating their use *in vivo*, and we have taken on the challenge of testing our best compound (MAG_{2P_F}^{slow}) in an animal model. Our results are shown in the new Fig. 6, and Supplementary Movies S5-S7. In order to describe and discuss them, we have added new paragraphs in the Results (new subsection in pages 19-20), Discussion (1st paragraph, page 23) and Methods (last paragraph, page 26) sections.

Briefly, we have carried out all-optical experiments with GluK2-L439C-mCherry and GCaMP6s fluorescent calcium reporter co-expressed in touch receptor neurons (TRNs) of *C. elegans* nematodes. From the six TRNs, which tile the receptive field of the animal in anterior/posterior and left/right, we specifically focused on the single pair PLML/R, which is located near the tail of the nematode (Fig 6a). Like in other neurons within the nervous system, although pairs of TRNs can be selectively stimulated in the anterior and posterior part of the animal using spatial light patterning, differential unilateral activation is difficult, which poses a hurdle to photomanipulate their activity at high resolution using optogenetics (Stirman et al, Nat. Methods 2011, Ref: 61). This is especially relevant when neurons are overlapping along the optical axis as in Fig. 6b, or are closely packed as in the head. Here, 2P excitation provides a unique advantage over 1P excitation to activate cells with axial plane selectivity of micrometers. In our experiments, MAG_{2P_F}^{slow} provided clear photoresponses to 2P excitation that were observed in every neuron and every animal responding to 1PE, leading to a 2PE/1PE efficacy of 100% (Fig. 6c), as previously found in brain slices (Fig. 5g; note that the efficacy of MAG is below 5%). Thus, the rationally designed 2P excitation properties of MAG_{2P_F}^{slow} can also be used *in vivo* to photomanipulate neuronal activity with high efficacy.

Our results solve in this way the question of how to control in three dimensions and with 100% reliability *in vivo* the activity of neuronal receptors. With the available pulsed lasers and holographic technologies, the transduction of light into biomolecular activity remains the bottleneck, and the question above comes down to the technical yet crucial question of how to design a photochromic group with high two-photon absorption cross section that can be integrated in potent and pharmacologically selective photoswitches.

Although *C. elegans* is a very important organism in genetics, metabolism, and neurobiology, and a model of choice to analyze neuronal circuits, the photocontrol of neuronal receptors and intracellular presynaptic proteins has been achieved very recently, and only using 1P excitation. Our first 2P excitation experiments *in vivo* with

MAG_{2P_F}^{slow} have been facilitated by genetic manipulation (including the expression of fluorescent calcium reporters), but the reliable photoresponses obtained open the way to directly introducing our novel *ortho*-fluorosubstituted azobenzene cores in freely diffusible photoswitches, in order to photocontrol endogenous receptors and signaling proteins with high spatiotemporal resolution.

In addition to the system application, very little is said about how the photoswitch actually works on glutamate receptors, or whether it alters the effects of the natural neurotransmitter glutamate on the receptor. The only functional measurements in the paper come from calcium imaging, a very indirect way to study and understand neurotransmitter receptors. Fundamental properties of the receptors, including apparent affinity for the probe (and for glutamate), activation and deactivation kinetics, and the extent and kinetics of desensitization, are not obtainable with optical imaging, and are not addressed in the paper.

We agree with the reviewer that functional effects of the photoswitch on the receptor are an important part of the characterization that we overlooked in our all-optical experiments. We combined optical stimulation and calcium fluorescence imaging in order to obtain significant statistics in large numbers of cells, and to increase the impact of our results in the chemical and biological communities due to the wide availability of two-photon fluorescence microscopes (e.g. in imaging facilities). However, electrophysiology indeed provides a more direct way to study neurotransmitter receptors, and we have performed new functional characterizations of our compounds using whole cell patch clamp of cells overexpressing GluK2-L439C (shown in Supplementary Figs. S6-S9 and Table S6).

The activation and deactivation kinetics of the receptor are very important for its performance in photoswitching experiments. We have quantified and compared the activation and deactivation lifetimes of GluK2-L439C conjugated to MAG, MAG_{2P}^{slow} and MAG_{2P_F}^{slow}. The results are shown in the new Supplementary Fig. S7 and Table S6, and indicate a less than twofold increase in T_{ON} and T_{OFF} of the new compounds compared to MAG under equal illumination conditions.

The apparent affinity of the MAG-conjugated receptor for glutamate and for the tethered ligand has been extensively characterized by one of the authors (Gorostiza et al, PNAS 2007, Ref: 37; Numano et al, PNAS 2009;) using both electrophysiology and calcium imaging. Due to the similarity of the ligand moieties of MAG, MAG_{2P}^{slow} and MAG_{2P_F}^{slow}, and the minor steric modifications caused by introducing a fluorine atom instead of hydrogen in the azobenzene moiety of MAG_{2P_F}^{slow}, their apparent affinities are expected to be very similar, and weaker than that of free glutamate. This fact allows the receptor conjugated to different MAG analogs to respond to illumination, while preserving the physiological responses to free glutamate. This was shown in Supplementary Fig. S10 and quantified in Supplementary Fig. S11, and we have performed new electrophysiological experiments that are summarized in the new Supplementary Fig. S8. Note that glutamate responses of the receptor conjugated to MAG, MAG_{2P}^{slow} and MAG_{2P_F}^{slow} are about 30% higher than the photoresponses, and that this difference strongly depends on the concentration of MAG analogs used during incubation (50 μM in Supplementary Figs. S10 and S11), as previously reported (Gorostiza et al, PNAS 2007, Ref: 37).

We have also carried out new patch clamp experiments in HEK293 cells with photoswitch-conjugated GluK2-L439C in order to examine the effect of desensitization (Supplementary Fig. S9). Steady-state photocurrents can be readily measured after cell incubation in concanavalin A (which blocks desensitization of kainate receptors) but prior to this treatment no photoresponses could be detected at 1 kHz sampling rate and 22 μW mm⁻² intensity of 380 nm illumination, indicating that full receptor desensitization occurs in less than few ms, and is not slowed down by photoswitch

conjugation. Physiological desensitization by ultrafast glutamate perfusion occurs in 3 ms and is unaltered by MAG conjugation and photoactivation with 100 μ s pulses at high illumination intensity at 5 mW mm⁻² (4 ms, Reiner and Isacoff, *Methods Mol. Biol.*, 2014; 1148:45-68). Detailed experiments to characterize the kinetics of desensitization and to quantify possible differences between MAG and our 2P-enhanced analogs are out of the scope of this paper, but the abovementioned similarity of the glutamate moiety in all these photoswitches suggests that they all preserve the desensitization properties of the receptor.

We have added an explanation in the main text to summarize these electrophysiological results (1st paragraph, page 12), and we have included two new references (Numano et al, *PNAS* 2009, 106:6814-6819; Reiner and Isacoff, *Methods Mol. Biol.*, 2014, 1148:45-68)

As it stands, the paper is very incomplete, and there is more work to do before a compelling case could be made for the new tool having a significant impact.

We think that the addition of new electrophysiological recordings to characterize the effect of the photoswitch on the responses to free glutamate of the receptor, and its activation and deactivation kinetics, as well as new all-optical experiments to demonstrate that two-photon control with MAG_{2P_F}^{slow} can be exerted *in vivo*, together have made the paper more complete. We hope that the chemical and biological impact of these novel photoswitches with enhanced 2P excitation responses has now been demonstrated convincingly. We are grateful to the reviewer for his/her positive criticisms and help to improve the quality and significance of the manuscript.

REVIEWERS' COMMENTS:

Reviewer #1 (Remarks to the Author):

The authors have revised their manuscript by including significant new data that clarify and expand their arguments.

I am happy with the changes and I support the manuscript publication.

Reviewer #2 (Remarks to the Author):

The authors have done a nice job of responding to my criticisms. I support publication.

Reviewer #3 (Remarks to the Author):

The reviewers have addressed my concerns and the revised paper has been improved significantly.